# Soil properties impacting denitrifier community size, structure, and activity in New Zealand dairy-grazed pasture.

Neha Jha[1, 2]*, Surinder Saggar[2], Donna Giltrap[2], Russ Tillman[1], and Julie Deslippe[2,a]

[1] Soil & Earth Sciences Group, Institute of Agriculture and Environment, Massey University, Palmerston North 4442, New Zealand
[2] Landcare Research, Palmerston North 4442, New Zealand
[a] Currently: School of Biological Sciences, Victoria University of Wellington, Wellington 6012, New Zealand

*Correspondence to*: Neha Jha (Neha_Jha@msn.com)

**Abstract.** Denitrification is an anaerobic respiration process that is the primary contributor of the nitrous oxide ($N_2O$) production from grassland soils. Our objective was to gain insight to the relationships between denitrifier community size, structure, and activity for a range of pasture soils. We collected 10 dairy pasture soils with contrasting soil textures, drainage classes, management strategies (effluent irrigated or non-irrigated), and geographic locations in New Zealand, and measured their physicochemical characteristics. We measured denitrifier abundance by quantitative polymerase chain reaction (qPCR) and assessed denitrifier diversity and community structure by terminal restriction fragment length polymorphism (T-RFLP) of the nitrite reductase (*nir*S, *nir*K) and $N_2O$ reductase (*nos*Z) genes. We quantified denitrifier enzyme activity (DEA) using acetylene inhibition technique. We asked whether varied soil conditions lead to different denitrifier communities in soils, and if so, whether they are associated with different denitrification activities and likely to generate different $N_2O$ emissions. Differences in the physicochemical characteristics of the soils were driven mainly by soil mineralogy and the management practices of the farms. We found that *nir*S and *nir*K communities were strongly structured along gradients of soil water and phosphorus (P) contents. By contrast, the size and structure of the *nos*Z community was unrelated to any of the measured soil characteristics. In soils with high water content the richnesses and abundances of *nir*S, *nir*K and *nos*Z genes were all significantly positively correlated with DEA. Our data suggest that management strategies to limit $N_2O$ emissions through denitrification are likely to be most important for dairy farms on fertile or allophanic soils during wetter periods. Finally, our data suggest that new techniques that would selectively target *nir*S denitrifiers may be the most effective for limiting $N_2O$ emissions through denitrification across a wide range of soil types.

## 1 Introduction

Nitrous oxide ($N_2O$), is a potent greenhouse gas that is produced as an intermediate product of biological nitrogen conversions in soils (Stevens et al., 1997). Denitrification is the stepwise anaerobic reduction of aqueous nitrate ($NO_3^-$) to nitrite ($NO_2^-$) and into the gaseous forms $N_2O$ and benign dinitrogen ($N_2$). It is the major global contributor to $N_2O$ production in grassland soils (Saggar et al., 2013) and is responsible for a significant fraction of agricultural greenhouse gas emissions (IPCC, 2014). Denitrification is mediated by the action of four enzymes: $NO_3^-$ reductase (NAR), $NO_2^-$ reductase (NIR), nitric oxide (NO)

reductase (NOR), and $N_2O$ reductase ($N_2OR$) (Zumft, 1997), which are encoded by the *nar/nap*, *nir*, *nor*, and *nos* genes, respectively. Taxonomically diverse bacteria, archaea (Philippot et al., 2007; Tiedje, 1994; Ishii et al., 2010) and eukaryotes (Zumft, 1997) are known to harbour two or more denitrification enzymes. Denitrifying bacteria are particularly widely distributed in pasture soils (Graham et al., 2014) and more than 60 genera have been identified (Chen et al., 2012). Denitrifiers with all four reductases are capable of emitting $N_2$ and are said to be 'complete' denitrifiers. Those denitrifiers that lack $N_2OR$ emit $N_2O$, as the final product of denitrification are called 'incomplete' denitrifiers.

*Nir*S, *nir*K, and *nos*Z genes have been targeted as functional markers of both complete and incomplete denitrifiers in soils (Stres et al., 2008; Throbäck et al., 2004; Morales et al., 2010; Enwall et al., 2010). The balance of complete and incomplete denitrifiers in soils can determine the ratio of $N_2O$: $N_2$ produced during denitrification (Philippot et al., 2011; Bakken et al., 2012). Thus, denitrifier community structure and abundance can be important factors in determining nitrogen (N)-loss and agricultural greenhouse gas emissions from soils. Indeed, a recent synthesis of 82 datasets relating bacterial community structure and environmental characteristics to a variety of carbon (C) and nitrogen (N) cycling processes found that microbial community structure data improved the power of models to explain denitrification process rates better than for any other ecosystem process (Graham et al., 2016). Still, strong relationships are not always observed between denitrification rates, and denitrifier community structure and abundance (Cavigelli and Robertson, 2000, 2001; Chèneby et al., 1998; Mergel et al., 2001). In particular, the structure of denitrifier communities in environmental samples is often poorly correlated with soil or environmental factors that are known to influence process rates (Dandie et al., 2011; Enwall et al., 2010; Philippot et al., 2009), indicating that our understanding of the factors controlling the diversity and function of denitrifying communities is still inadequate. Moreover, there is a need to identify the soil conditions in which the presence and activity of denitrifiers are likely to lead to substantial $N_2O$ emissions, so that appropriate strategies for targeted and effective management can be deployed or developed where they are lacking.

Pastoral dairy farming is a preferred land use in mild and wet climates on relatively fertile soils and flat sites that occupy low-lying positions in the landscape as these locations support high rates of pasture production (Saggar et al., 2013). The combination of periodically anoxic soil conditions, high concentrations of N in cattle excrement patches and relatively high microbial biomass at these sites, combine to favour denitrification as a major oxidative metabolic pathway. Despite this, denitrification rates and potentials as well as $N_2O$ emissions through denitrification vary widely among pasture soils (Cayuela et al., 2013; Giltrap et al., 2011; Groffman et al., 2006). Soil management practices including the addition of organic amendments such as plant residues, compost, manure, or effluent irrigation can increase soil fertility and microbial biomass, and may lead to structural shifts in soil microbial communities, which in turn alter soil biochemical processes (Kennedy and Smith, 1995). The addition of

crop residues to soils increases the abundance of denitrifier genes and leads to greater denitrification in soils (Barrett et al., 2016; Gao et al., 2016; Henderson et al., 2010). Likewise, increasing soil water content is associated with increasing denitrifier gene abundances in soils (Liu et al., 2012; Mergel et al., 2001). Management practices that alter the size of the denitrifier community in soils are also likely to affect its denitrification enzyme activity (DEA), as the abundance of denitrifier genes can be a strong determinant of DEA (Čuhel et al., 2010; Deslippe et al., 2014; Enwall et al., 2010; Hallin et al., 2009). However, the geologic origins of a soil can determine its dominant properties over a range of soil C and water contents (Bronick and Lal, 2005). Indeed, we previously found that soil texture, drainage class, and latitude were powerful regulators of denitrification end products ($N_2$ vs. $N_2O$) and total emissions, also both the forms and quantity of gases emitted could be predicted by the 16S rRNA gene communities of soil samples (Morales et al., 2015). However, we still lack detailed knowledge of how variation in soil properties affect denitrifier populations and denitrification. Better information on the role of soil physicochemical characteristics in determining the size and activity of denitrifiers may allow for improved and soil-specific management of $N_2O$ emissions from pastoral agriculture.

Here, we sought a better understanding of the relationships between the structure, abundance, and activity of denitrifiers over a range of New Zealand dairy-pasture soils, which varied widely in soil properties and had different management conditions. We asked if the properties of these soils drove unique denitrifier communities that supported different DEA or were likely to generate different $N_2O$ emissions. We expected to find that the size and structure of denitrifier communities' would vary most strongly in accordance with soil water content and that soil physical properties or management practices that increase soil water would enhance the size and activity of denitrifiers.

## 2 Materials and Methods

### 2.1 Sites and soils

Our aim was to sample soils that would encompass the range of physicochemical conditions that predominate on New Zealand dairy farms. We therefore targeted soils on the basis of their geographical location (North or South Island of New Zealand), and mineralogy (allophanic or non-allophanic soils). As soil water content is a key factor affecting the structure and activity of soil denitrifier communities (Liu et al., 2012; Mergel et al., 2001) it was also important to sample in both wet and dry seasons. We therefore sampled soils over a 6-month period from winter to summer. Soil textures varied from a stony silt loam to a fine sandy loam, and the sites ranged from poorly drained to well drained (Table 1). We sampled soils expected to have the greatest soil water contents in winter and those we expect to be driest in summer, with other soils sampled in between these times (see supplementary table 1 for soil sampling dates). We collected soils from 10 different commercial dairy farms (Fig. S1). All were fenced from

livestock and none had been grazed within 8 weeks of sampling. All sites were dominated by perennial ryegrass (*Lolium perenne)* and white clover (*Trifolium repens*). Fertilization regimes varied among the farms and consisted of applications of 150–200 kg N ha$^{-1}$ annually. Detailed descriptions of the individual fertiliser applications at the 10 farms are described in the supplementary information.

*Insert Table 1*

## 2.2. Sampling and analysis of soil properties

At each farm, we randomly selected six blocks of 100 m$^2$ for the collection of soil samples. At randomly selected locations within each block, twenty–five soil cores (25 mm diameter × 100 mm long) were obtained using a steel corer. The 25 cores from each block were pooled to form a single composited

sample per block ($n = 6$ composited soil samples per farm). All soil samples were collected between August and December 2010 once from each site. Soil samples were taken to the laboratory, individually homogenised, sieved to 2 mm and stored at 4ºC in plastic bags (10 sites × 6 replicates = 60 samples). A sub-sample of each soil was stored at –20ºC for molecular analysis. We measured pH, nitrate (NO$_3^-$) & ammonium (NH$_4^+$) –N (mineral-N), total nitrogen (TN), total carbon (TC), Olsen phosphorus (P), and

soluble C, on the field-moist sieved soils using standard protocols (for details see Jha, 2015). All soils were analysed for these parameters within 2 weeks of sampling.

DEA was determined using the acetylene inhibition method described in Luo et al. (1999), with the exception that we added chloramphenicol to inhibit the *de novo* synthesis of enzymes. Thus the values

we report represent only the existing enzyme activity in soils. DEA was assessed for all soil samples within 2 days of collection. DEA incubation conditions and the gas sampling methods are described elsewhere (Jha, 2015). We intended to measure microbial biomass carbon (MBC) within 48 hours of soil sampling but a technical problem with our set-up delayed measurements of MBC for nearly 3 months. To standardize this effect across soil samples we monitored changes in the size of the MBC pool in two soils

over 7 months. We found that no significant changes in MBC occurred between 3 and 7 months for soils stored at 4ºC (see supplementary table 1). We therefore report MBC data for all soils that were stored at 4ºC between 4 and 6 months.

## 2.3 DNA extraction from soils

Within six months of soil sample collection, soil samples were thawed on ice and a 0.25 g aliquot was obtained. DNA was extracted from these soil samples using the MoBio PowerSoil™ DNA Isolation Kit (MoBio, Solana Beach, CA, USA) following the manufacturer's instructions. The yield and quality of DNA extracts were verified as described in Deslippe et al. (2014). DNA was stored at –20ºC until analysed.

## 2.4 Terminal restriction fragment length polymorphism (T–RFLP) of denitrifier genes

Terminal restriction fragment length polymorphism (T–RFLP) was performed to analyse the community structure and diversity of *nir* and *nos* genes in soil samples. T-RFLP for *nir*S and *nos*Z genes was conducted as described in Deslippe et al. (2014) except that PCR for *nir* genes occurred in a total volume of 25 µl reaction mixture, which contained 2.5 µl of 10×PCR buffer (1 mM MgCl$_2$), 0.5 mM MgCl$_2$, 0.2 mM each deoxynucleotide triphosphate (dNTP), 1.25 U of *Taq* polymerase (Fisher *Taq*, Thermofisher Scientific® Inc.), 0.8 mg/ml Bovine Serum Albumin (BSA), 1.0 µM of each primer, and 10 ng DNA template per reaction. The PCR amplification consisted of an initial denaturation of the DNA template at 94$^o$C for 30 s, followed by 35 cycles of 20 s at 94$^o$C, 20 s at 56$^o$C, and 20 s at 68$^o$C. The reaction was completed by 10 min at 68$^o$C.

For T-RFLP of the *nir*K gene we used the primers Copper 583F, 909R (Dandie et al., 2011). The amplifications of *nir*K and *nos*Z genes were achieved under slightly different condition than the *nir*S gene according to the specifications of the reagents used for PCR. The PCR amplification was performed in a total volume of 25-µl reaction mixture containing 10 µl of $2 \times$ NEB Taq master mixes (New England Biolabs® Inc.), 0.4 µM of each primer, and 10 ng DNA template per reaction. PCR consisted of an initial denaturation of the DNA template at 94$^o$C for 2 min, followed by 35 cycles of 30 s at 94$^o$C, 1 min at 56$^o$C, and 1 min at 72$^o$C. The reaction was considered complete after 10 min at 72$^o$C.

The T-RFLP profiles generated for the soil samples were analysed using Peak Scanner® v1 software (Life Technologies) and as described in Deslippe et al. (2014). The total number of terminal restriction fragments (T-RFs) per electropherogramme was taken to indicate genotype richness per sample. We then calculated the gene Shannon's diversity index and Pielou's evenness index (Magurran, 1988) per sample and used 1-way analysis of variance (ANOVA) to determine if soils belonging to the three physicochemical groups differed with respect to gene richness, evenness and diversity.

## 2.5 Quantitative polymerase chain reaction (qPCR) of total bacterial and denitrifier genes

Quantification of bacterial *nir*S, *nir*K, and *nos*Z genes was accomplished using qPCR, following the methodology of Deslippe et al. (2014). Amplification efficiencies of qPCR reactions for samples were within the range of values (E = 90–110%) published previously (McPherson and Moller, 2006). The reactions were linear over 7 orders of magnitude and sensitive down to $10^2$ copies. The ratio of abundances of denitrifier genes in environmental samples has been interpreted previously as an index of the potential for complete denitrification (Philippot et al., 2009). Here, we calculated the *nos*Z:(*nir*S+*nir*K) of soil samples. We expected that soils with low *nos*Z:(*nir*S+*nir*K) are more likely to emit a greater N$_2$O:(N$_2$O + N$_2$).

## 2.6 Statistical Analysis

The normality and homoscedasticity of all soil physicochemical, gaseous emission and biological datasets were examined using Anderson-Darling (Stephens, 1986) and Levene's tests, respectively in Minitab® 16 software (Minitab Inc.). Box-Cox transformations (Box and Cox, 1964) were applied to data sets as required to conform to model. The differences in the means of soil characteristics such as pH, nitrate ($NO_3^-$) & ammonium ($NH_4^+$) –N (mineral-N), total nitrogen (TN), total carbon (TC), Olsen phosphorus (P), microbial biomass carbon (MBC), soluble C, DEA, number of gene T–RFs and gene copy numbers were assessed using a one-way analysis of variance (ANOVA) test with soil type as a factor. Tukey's Studentized Range Test at $\alpha = 0.05$ significance level was used *post hoc* to reveal significant differences among means. The relationships among the soil chemical characteristics pH, nitrate ($NO_3^-$) & ammonium ($NH_4^+$) –N, TN, TC, Olsen P, MBC, DEA, number of denitrifier gene T–RFs, and gene copy numbers were determined using Pearson's correlation analysis.

In order to reduce the dimensionality of the many correlated soil physicochemical characteristics we performed principal components analysis (PCA). We included % soil water content (SWC), % SWC at field capacity (% FC SWC), pH, TN, TC, Soluble C, Olsen P, nitrate ($NO_3^-$) and ammonium ($NH_4^+$) –N as factors in the PCA. Soils grouped along the first and second ordination axes. We used multiple response permutation procedure (MRPP) to assess the statistical significance of these groupings. MRPP calculates the chance-corrected within-group agreement (A), a measure of within-group homogeneity compared with that expected by chance, where A = 1 corresponds to identical members within each given group (maximum effect of factor), and where A ≤ 0 corresponds to within-group heterogeneity equal to or larger than that expected by chance (no effect of factor; McCune and Medford 1999). We also calculated Pearson correlations among soil microbial characteristics and the ordination axes, and plotted those that were significantly correlated (tau>0.2) with axis 1 and 2 as vectors on the PCA.

Analysis of the *nir*S, *nir*K and *nos*Z community structure was based on threshold normalised peak heights of T-RFs from electropherogrammes (Deslippe et al., 2014). Non-metric multidimensional scaling (NMS) ordinations were performed using Bray & Curtis distance (Bray and Curtis, 1957) in the programme PC-ORD (McCune and Mefford, 1999). In order to illustrate how the structure of denitrifier communities varied with the physicochemical characteristic of soils, we calculated Kendall's rank correlations among physicochemical and biological characteristics of soils with the NMS ordination axes in PC-ORD. The significant correlates (tau > 0.2) were overlaid as vectors on the NMS ordination plots.

# 3 Results

## 3.1 Variations in soil chemical characteristics

The 10 soils differed significantly with regard to all measured physicochemical characteristics (Table S2). The PCA of soil characteristics generated 3 significant axes, of which the first two accounted for 83.4 % of the total variance (Fig. 1). Axis 1, which accounted for 53.4% of the variation in soil properties, described primarily a difference in % FC SWC, although total N and total C, and Olsen P also weighed heavily in forming Axis 1. Axis 2, which accounted for 30.0% of the variation in soil properties, described primarily a gradient in mineral N-form, with $NO_3^-$–N increasing and $NH_4^+$–N decreasing along Axis 2. The 10 soils segregated into three groups, with replicates of a soil tending to cluster closely together in the PCA. Firstly, the two allophanic soils OH (Otorohanga silt loam) and HR (Horotiu silt loam) (group 1) were separated from all other soils by their relatively high % FC SWC, their high total N, C and allophane contents. Secondly, the effluent-irrigated soil, MWEI (Manawatu effluent-irrigated fine sandy loam), was separated from all other soils (group 2) due to its high $NO_3^-$–N content. The seven remaining soils formed a loose cluster due to their relatively high $NH_4^+$–N and low Olsen P contents (group 3). MRPP confirmed that these groups differed significantly in soil physicochemical characteristics (A= 0.379, $P < 0.001$, Fig. 1). Table S3 summarises the physicochemical characteristics for the 3 groups of soils. Overall, we found that axes 1 and 3 of the PCA were not significantly correlated to DEA or MBC. However, axis 2 of the PCA, which describes a gradient in mineral N-form, was significantly positively correlated to DEA ($r^2 = 0.214$) and to MBC ($r^2 = 0.303$; Fig. 1).

*Insert Figure 1*

## 3.2 Bacterial denitrifiers in New Zealand dairy-grazed pasture soils

### 3.2.1 Diversity indices of denitrifier gene T-RF profiles

Across all soil samples, total T-RF richnesses were *nir*S=52, *nir*K=53, *nos*Z=47, which are typical values for T-RFLP studies of functional genes in soils (Deslippe et al., 2014; Rich and Myrold, 2004; Rösch and Bothe, 2005).While the minimum and maximum numbers of T-RFs varied among samples, the patterns of richness, evenness, and diversity among soils belonging to the three groups of soils were similar (see supplementary table S4), for simplicity, we therefore present only the values for T-RF richness in figure 2. *Nir*S communities among the three groups of soils had similar richnesses. By contrast, the allophanic soils of group 1 had significantly lower *nir*K richness, while the effluent irrigated soils of group 2 had significantly greater richness than all other soils (Kruskal-Wallis H = 13.84, $P = 0.0001$). *Nos*Z richness was significantly greater in the effluent irrigated soils of group 1 compared with all other soils (Kruskal-Wallis H = 8.59, $P = 0.014$).

*Insert Figure 2*

### 3.2.2 Denitrifier community structure in soils

Ordination of the soil samples in *nirS* T–RF space indicated significant structuring of the *nir*S community according to variation in the physicochemical characteristics of the soils (Fig. 3). Interestingly however, the variation in *nir*S community structure was not driven by the same physicochemical characteristics that varied most widely among soils and formed the first and second PCA axes. Consequently the *nir*S communities of soil samples did not cluster according to the three groups of soils in our PCA. Rather, Axis 1 which accounted for 30.0% of the variation in *nir*S community structure, was significantly correlated to SWC and the Olsen P contents of the soils (Fig. 3). This indicates that *nir*S community structure in dairy pasture soils from across New Zealand responded most strongly to SWC and P gradients. Likewise, 66.9% of the variation in *nir*K community structure (axis 1) was significantly correlated to the SWC and Olsen P contents of the soils. Axis 1 of the *nir*K ordination primarily separated two soils (Horotiu silt loam, and Paparua Lincoln silt loam) from all other soils. However, even when these two soils were removed from the dataset, NMS ordination revealed that the *nir*K community was primarily structured according to Olsen P and soil water variables (both SWC and % FC SWC; supplementary fig S2A). An additional 7.7% of the variation in *nirK* community structure was most strongly correlated to *nir* gene abundance (*nir*S + *nir*K), in soils, which was higher in the HR soil, an allophanic soil, than for any other. Ordination of the soil samples in *nosZ* T-RF space revealed little clustering of soil samples by origin or group. Likewise, we detected no significant patterns of correlation among the first and second ordination axes and the soil physicochemical characteristics. However axis 1 of the NMS ordination was also most strongly correlated to SWC and Olsen P (supplementary fig S2B).

*Insert Figure 3*

### 3.2.3 Denitrifier gene abundance

The number of *nir*S and *nir*K gene copies varied widely among the 10 soils; *nir*S gene copies ranged from $2.5 \times 10^7$ to $3.9 \times 10^8$ copies g$^{-1}$ soil, while *nir*K gene copies varied from $2.3 \times 10^8$ to $5.9 \times 10^8$ g$^{-1}$ soil. Overall soils, *nir* genes were on average an order of magnitude more abundant than those encoding the final step of denitrification. *Nos*Z gene copies varied most widely ($7.1 \times 10^6$ to $4.8 \times 10^7$ g$^{-1}$ soil), a much greater range than for *nir*S+*nir*K gene copies. The sum of *nir* gene (*nir*S+*K)* copies was significantly greater in the allophanic soils of group 1 than in the soils of group 3 ($P<0.005$, Fig. 4a), with the effluent irrigated (group 2) soil having intermediate values. Despite large variability, the group 2 soil had significantly more *nos*Z gene copies than the soils in group 1 and 3, while the group 1 soil had significantly fewer than the other two groups ($P<0.005$, Fig. 4a). Consequently the ratio of *nosZ* : (*nir*S+*K)* genes, which may indicate of the relative abundance of complete denitrifiers, varied significantly among the three groups of soils (Fig. 4b), with the effluent irrigated group 2 soil harbouring the highest, the allophanic soils of group 1 the lowest and the group 3 soils intermediate ratios of *nos:nir* genes.

**3.3 Denitrification enzyme activity (DEA)**

DEA varied considerably among the pasture soils but also among replicates within a soil (Table S3). DEA varied by a factor of five among the soil groups, with the group 2 soil (effluent irrigated MWEI) achieving significantly higher DEA than other groups, while the soils of group 1 (HR & OH) had significantly lower DEA values than other groups (H = 12.09, *P* = 0.02; Table S3). DEA varied considerably in soils belonging to group 3. Overall, DEA was mostly strongly positively correlated soil $NO_3^-$–N contents and was mostly strongly negatively correlated to soil $NH_4^+$–N contents, driving its significant correlation with axis 2 of our PCA (Fig 1). All significant physicochemical correlates to DEA are given in Table S5.

**3.4 Relationships among denitrification and denitrifier community size and structure across a range of soil moistures**

Given that the structure of *nir*S, *nir*K and *nos*Z communities varied primarily in response to soil water content and Olsen P (Fig 3), we wished to know if unique relationships between the richness and size of the denitrifier gene community and DEA exist at different SWCs. To address this question, we categorized soils according to coarse-scale SWC (high, moderate and low) and examined Pearson's correlations among these variables within soil SWC categories. For soils in the highest SWC category (MWEI, OH, and HR), we found that strong and significant positive correlations existed between denitrifier gene copy numbers and DEA [*nos*Z (r = 0.643, *P* = 0.049), *nir*K (r = 0.821, *P* = 0.007)]. Likewise, strong and significant positive correlations existed between DEA and the T-RF richness of denitrifier genes [*nos*Z (r = 0.801, *P* = 0.010), *nir*K (r = 0.783, *P* = 0.013), and *nir*S (r = 0.793, *P* = 0.011)]. However, these patterns of correlation were not present in the soils categorized as moderate or low SWC.

**4 Discussion**

Despite its relatively small total land area, New Zealand is geologically diverse and the 1.8 million hectares of land that were managed as dairy pasture in 2015 (Dairy NZ, 2017), have soils derived from a wide range of parent materials. Here we studied 10 dairy pasture soils that varied widely in texture, drainage class and management strategies. We found that the % FC SWC and a gradient in mineral N-form accounted for the greatest variation in soil physicochemical characteristics, and that key microbial parameters for denitrification such as MBC and DEA were significantly positively correlated with higher soil $NO_3^-$–N. In our study, these patterns were driven primarily by only three soils; the two allophanic soils, which had high % FC SWC (group 1), and the effluent irrigated soil, which had very high $NO_3^-$–N (group 2). The effluent irrigated soil, which had the highest MBC, likely harboured a larger population of nitrifiers, whose activities generated the $NO_3^-$ required by denitrifiers, and supported the highest rates of DEA we observed. Nonetheless, our results are consistent with previous

reports that soil microbial biomass is a key indicator of denitrification process rates (Drury et al., 1991). From this perspective, across a wide range of soil properties, the size of the MBC pool may be an important coarse-scale indicator of soil $N_2O$ emissions under both anoxic (denitrification) and oxic (nitrification) conditions.

Allophanic soils have high water content at field capacity, but they adsorb copper, and are therefore likely to select against *nir*K denitrifiers, whose periplasmic nitrite reductase requires six copper atoms to maintain its trimeric structure. This was reflected in our data by the very low richness, evenness and diversity of *nir*K T-RFs, as well as in the very low numbers of *nir*K gene copies, relative to the other soils. We expected this to also reduce the overall number of genes encoding nitrite reductase in group 1 soils, but didn't observe this. Instead we found that *nir*S denitrifiers replaced *nir*K denitrifiers in allophanic soils, so that the total number of *nir* gene copies was equivalent to that in the effluent irrigated soil and significantly greater than the number of *nir* copies in all other soils. Interestingly, despite the large size of the *nir*S community, allophanic soils did not, on average, have more diverse *nir*S communities than other soils. However, the size and diversity of *nir*S communities in allophanic soils was more variable than for other soils. These findings suggest that allophanic soils support relatively few microsites where denitrification, driven by *nir*S denitrifiers, is the dominant respiratory pathway. New Zealand's allophanic soils are porous and free-draining with relatively low bulk densities (Molloy, 1998). As such, anoxic microsites conducive to denitrification are expected to be few. Likewise, fewer active microsites for denitrification fits with the low to moderate DEA we observed in the allophanic soils. Allophanic soils are known to adsorb P (Hashizume and Theng, 2007), and the binding of adenosine by allophanes may have limited DEA in these soils despite their relatively large *nir* populations. Nonetheless, we also found far fewer copies of *nos*Z genes, relatively low *nos*Z diversities, and the lowest *nos:nir* gene ratios in the allophanic soils, suggesting that complete denitrifiers are relatively rare in these soils. Consequently, where and when it occurs, denitrification in allophanic soils is likely to lead to significant $N_2O$ emissions. This result fits with other work from our group, which indicates that allophanic soils emit greater $N_2O:(N_2O + N_2)$ relative to other soil types (McMillan et al., 2016). Taken together, these results suggest that targeted management of *nir*S denitrifiers in allophanic soils during wet seasons may be an effective strategy to combat greenhouse gas emissions from pastoral agriculture in volcanic regions.

The effluent irrigated soil (MWEI), with physicochemical properties that separated it from all other soils, was characterised by very high $NO_3$ and Olsen P concentrations, relatively high pH (5.9) and high MBC, which supported very high DEA. This moderately drained, fine sandy loam had the highest SWC at the time of sampling. MWEI had the largest number of *nir*K gene copies, but only moderate numbers of *nir*S, leading to intermediate total numbers of *nir* genes. Likewise, it had the greatest diversity of *nir*K genotypes, but only moderate diversity of *nir*S genotypes. These findings emphasise the potential for

effluent irrigation to increase denitrification enzyme activity, likely through increasing both the size of the total microbial community (MBC), SWC and $NO_3$ availability, which in turn selects for denitrifiers. However, MWEI supported a significantly larger population of *nos*Z denitrifiers than the other soils and this led to the highest *nos:nir* of any soil. Overall, these findings suggest that MWEI is likely to support a large and active community of denitrifiers but that complete denitrification may limit $N_2O$ emissions from this soil. Consequently, management of greenhouse gas emissions from highly fertile pasture soils like MWEI may benefit from strategies that limit $NO_3$ availability in soils, such as the application of nitrification inhibitors (e.g. DCD).

When considered in isolation, the seven soils of group 3 still varied significantly with regard to physicochemical characteristics. In particular MW, PS and PL, which ranked higher on Axis 2 of the PCA, differed from the four remaining soils (Fig.1). For example, they had on average three times the $NO_3$ and half the $NH_4^+$ as the other group 3 soils. These soils also had the three highest pH values (6.0-6.4), and high MBC, but relatively low SWCs. They supported relatively large numbers of *nir*S and *nos*Z denitrifiers, but only average numbers of *nir*K genes, leading to overall intermediate *nir:nos*. Likewise, these soils had intermediate diversities of *nir*S, *nir*K and *nos*Z genes and ordinations of the *nir*S, *nir*K and *nos*Z gene T-RFs, failed to distinguish these soils from those in the other groups. Despite this when incubated under non-limiting conditions, these three soils, together with MWEI, supported the highest DEAs. These findings indicate that denitrification responds quickly to SWC in moderately fertile soils. Thus, careful management of $NO_3^-$ loads by limiting dairy stock or the use of nitrification inhibitors or both, is also likely to be useful to limit greenhouse gas emissions from these soils during wetter periods of the year.

The four remaining soils of group 3 were the least fertile and the most acidic (pH-4.8-5.7) with the lowest MBC in our study. They were also among the driest. Despite moderately high total numbers of *nir*S, *nir*K and *nos*Z genes, the *nos:nir* in these soils were equivalent to the other soils of group 3 and intermediate overall. With the exception of the two allophanic soils, the four remaining soils of group 3 had the lowest DEA, which was on average about a quarter of that measured in the other group 3 soils. Taken together, these results suggest lower risk of $N_2O$ emissions from these soils, as oxic conditions and low concentrations of substrates are likely to limit denitrification much of the time, and moderately high numbers of *nos*Z denitrifiers will favour some complete denitrification of this smaller total N pool.

Overall, ordinations of T-RFLP data revealed no structuring of the *nir*S, *nir*K or *nos*Z communities according the three groups that defined the major physicochemical characteristics of the soils, a gradient in soil water content at field capacity and a gradient in mineral N forms. Rather, SWC at the time of soil sample collection and Olsen P were the primary drivers of the structure of denitrifier communities. Given the high correlation of SWC and Olsen P overall in soil samples, this result is likely to indicate

considerable plasticity of the denitrifier community in response to ambient soil water content. In the wettest soils, we found strong and significant positive correlations between the diversity of *nir*S, *nir*K and *nos*Z genes and DEA. We also found strong and significant positive correlations between *nir*K and *nos*Z gene copy numbers and DEA in those soils. However, these relationships broke down for soils with moderate or low SWCs. While the aim of our study was to sample pasture soils over a wide range of physicochemical characteristics in order to gain insight to the properties of the denitrifier communities they support, seasonal variation in the structure of *nir*S and *nir*K denitrifiers in cultivated and pasture soils is well established (Wertz et al., 2016; Tatti et al., 2017; Bent et al., 2016; Yu et al., 2016; Smith et al., 2010). The plasticity of the denitrifier community in response to ambient soil water content, together with the strong correlations between the size, diversity and activity of denitrifying communities in very wet soils suggests that future work towards characterising the denitrifier communities most likely to contribute to greenhouse gas emissions from pastoral soils should focus sampling efforts on the wettest times of the year.

Several lines of evidence collected here suggest that *nir*K denitrifiers were more sensitive to the range of physicochemical characteristics in soils than were *nir*S denitrifiers. For example, gene copy numbers and diversity metrics of *nir*S communities were fairly uniform across soil groups, but varied significantly for *nir*K denitrifiers. Likewise, the patterns of *nir*K diversity across soil groups were mirrored by the patterns of *nos*:*nir*, suggesting that changes in the size of the *nir*K community had a dominant influence on the overall ratio of complete and incomplete denitrifiers in soil groups. Likewise, independent shifts in *nir*S and *nir*K community structures in response to common physicochemical characteristics was recently observed in a eutrophic reservoir (Zhou et al., 2016). In contrast, the structure of *nos*Z communities did not correspond to any physicochemical property measured. Together, these results may suggest that *nir*S and *nos*Z genotypes are equivalently adapted to the physicochemical conditions of a wide range of dairy pasture soils, while *nir*K denitrifiers are more sensitive. Given that our data suggests greater $N_2O$ emissions in allophanic soils where *nir*K denitrifiers are few, it may be the microbial communities dominated by *nir*S denitrifiers should be the target of efforts to reduce greenhouse gas emissions from pasture soils. However, further work is necessary to confirm whether microbial communities dominated by *nir*S denitrifiers support greater $N_2O$ emissions than *nir*K denitrifier communities of equivalent size.

**5 Conclusions**

Here we characterise the size, structure and diversity of *nir*S, *nir*K and *nos*Z genes in soils that varied widely in physicochemical characteristics to address the question of whether different denitrifier communities develop under these varied soil conditions, and if so, whether they are associated with different denitrification activities and likely to generate different $N_2O$ emissions. Overall, we found a

strong correlation between MBC and DEA and that moderately high to highly fertile soils supported the largest populations of denitrifiers. Given that the more fertile soils were also likely to harbour significant populations of nitrifiers MBC may be an important coarse-scale indicator of total potential $N_2O$ emissions from such soils. However, our results for allophanic soils suggest that even relatively low rates of denitrification may lead to significant $N_2O$ emissions given their relatively low *nos:nir*. Consequently, we conclude that management strategies to limit $N_2O$ emissions through denitrification are likely to be most important for dairy farms on fertile or allophanic soils during wetter periods. Finally, our data suggest that new techniques that would selectively target *nir*S denitrifiers may be the most effective for limiting $N_2O$ emissions through denitrification across a wide range of soil types.

## 6 Acknowledgements

We acknowledge the funding and support for this study by New Zealand Agricultural Greenhouse Gas Research Centre (NZAGRC), through Landcare Research's Strategic Science Investment Funding from the Ministry of Business, Innovation and Employment (MBIE). We also acknowledge support from Peter Berben & Thilak Palmada, Landcare Research, Palmerston North; Jiafa Luo, AgResearch, Hamilton; and Mohammad Zaman, Balance Agrinutrients, for assistance and arrangement for collection of soil samples from Manawatu, Waikato, Christchurch, and Ashburton.

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

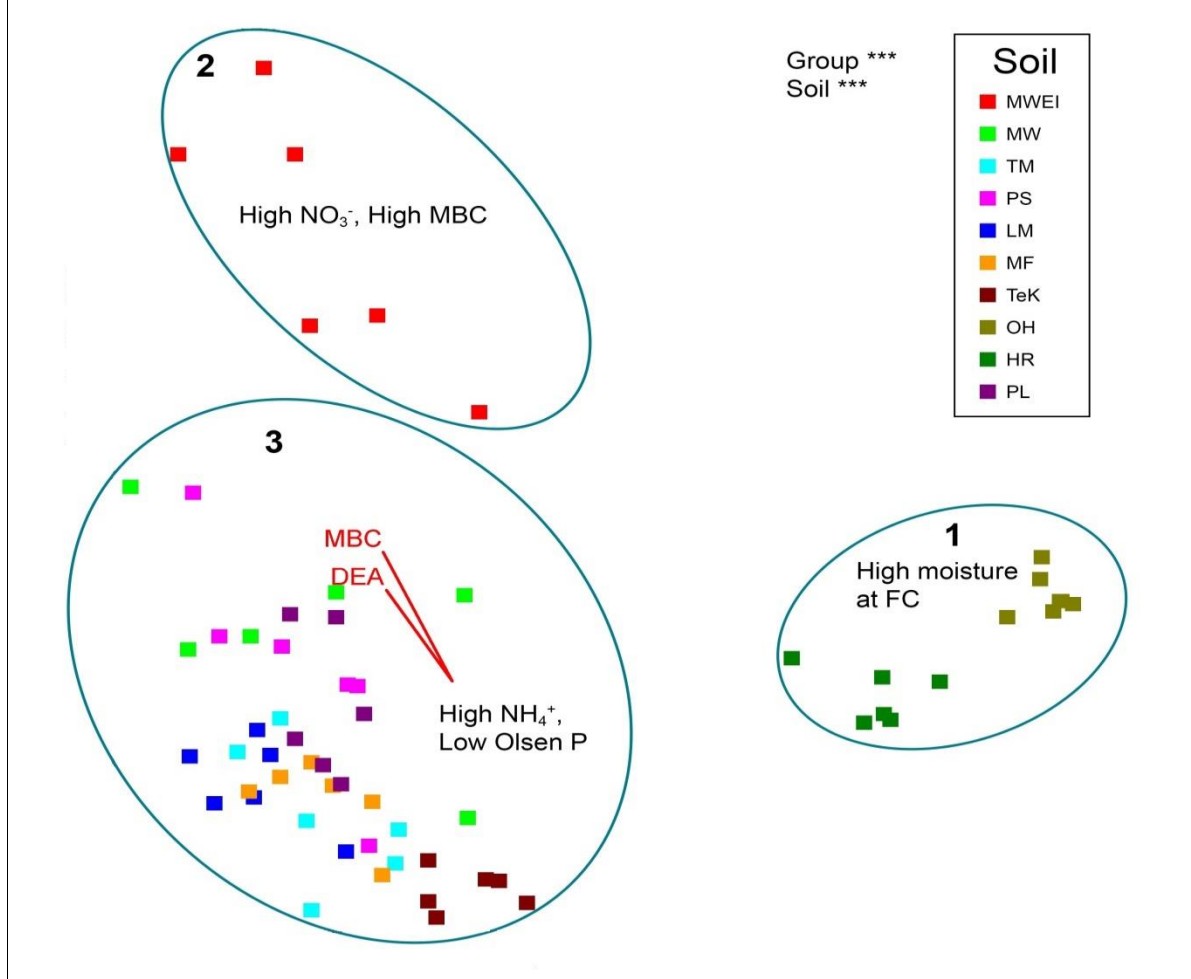

**Fig. 1** Principal component's analysis of the physicochemical characteristics of soil samples collected at
10 dairy farms and results of multiple response permutation procedure (MRPP) to assess the significance
of soil origin and group. Vectors indicate significant correlates (tau>0.2) with ordination axes in 1st and
2nd PC axes.

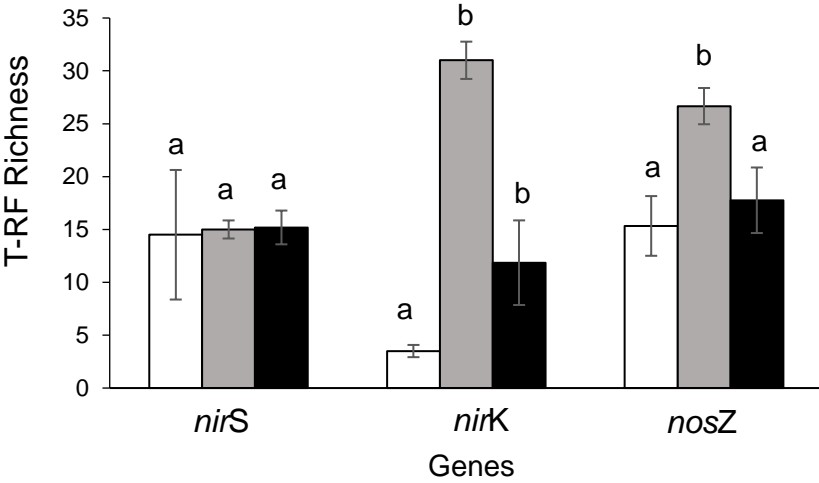

5    **Fig. 2** T-RF richness by soil group. White, grey and black bars denote groups I, II and III soils. Mean

values are reported ± 1 standard error of the mean. Means with the same letters do not differ significantly

at alpha=0.5

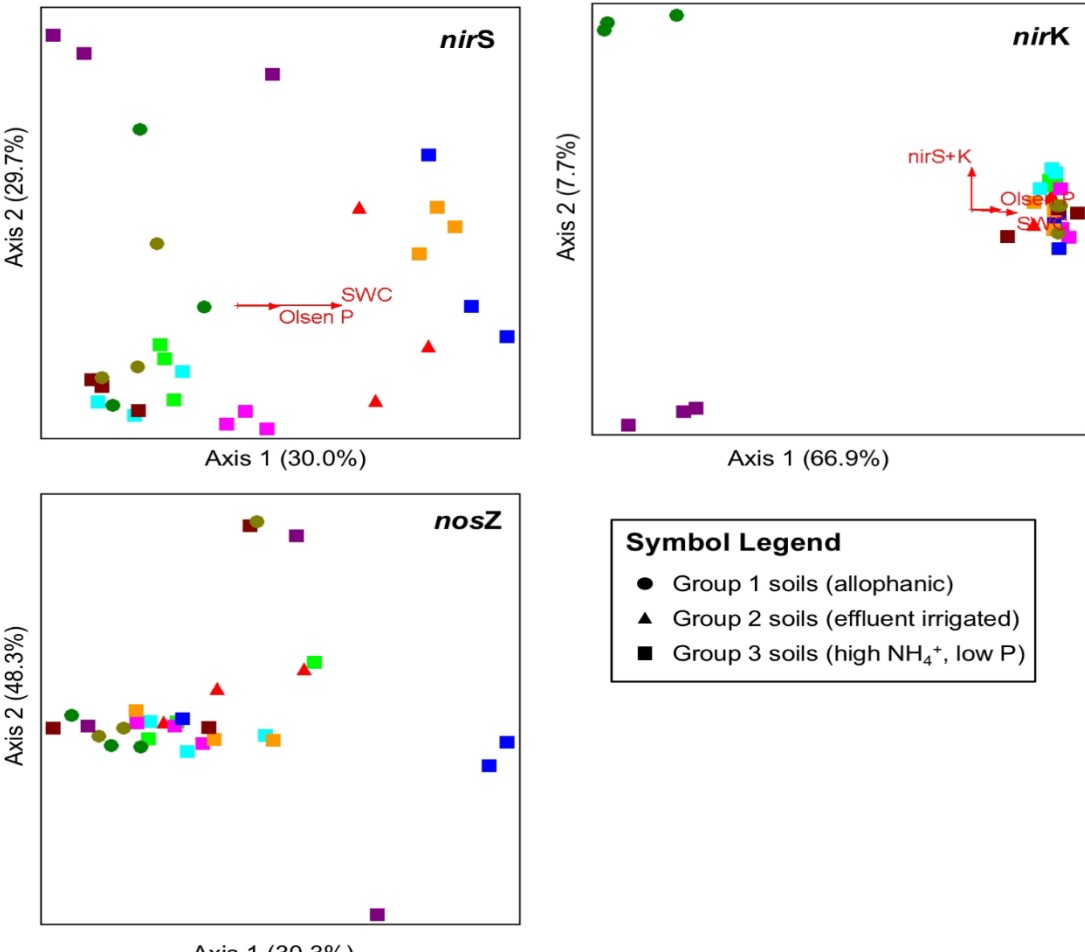

**Fig. 3** Non-metric multidimensional scaling (NMS) ordinations illustrating the Bray-Curtis dissimilarities of *nir*S, *nir*K and *nos*Z communities. Vectors represent those factors that were significantly correlated to the first and second ordination axes at tau = 0.2. Symbol colours are the same as in Fig. 1. Symbol shapes represent the three groups of soils as determined by PCA of their physiochemical characteristics (Fig 1). Final stress values for ordinations: *nir*S 12.53, *nir*K 5.58, *nos*Z 9.46.

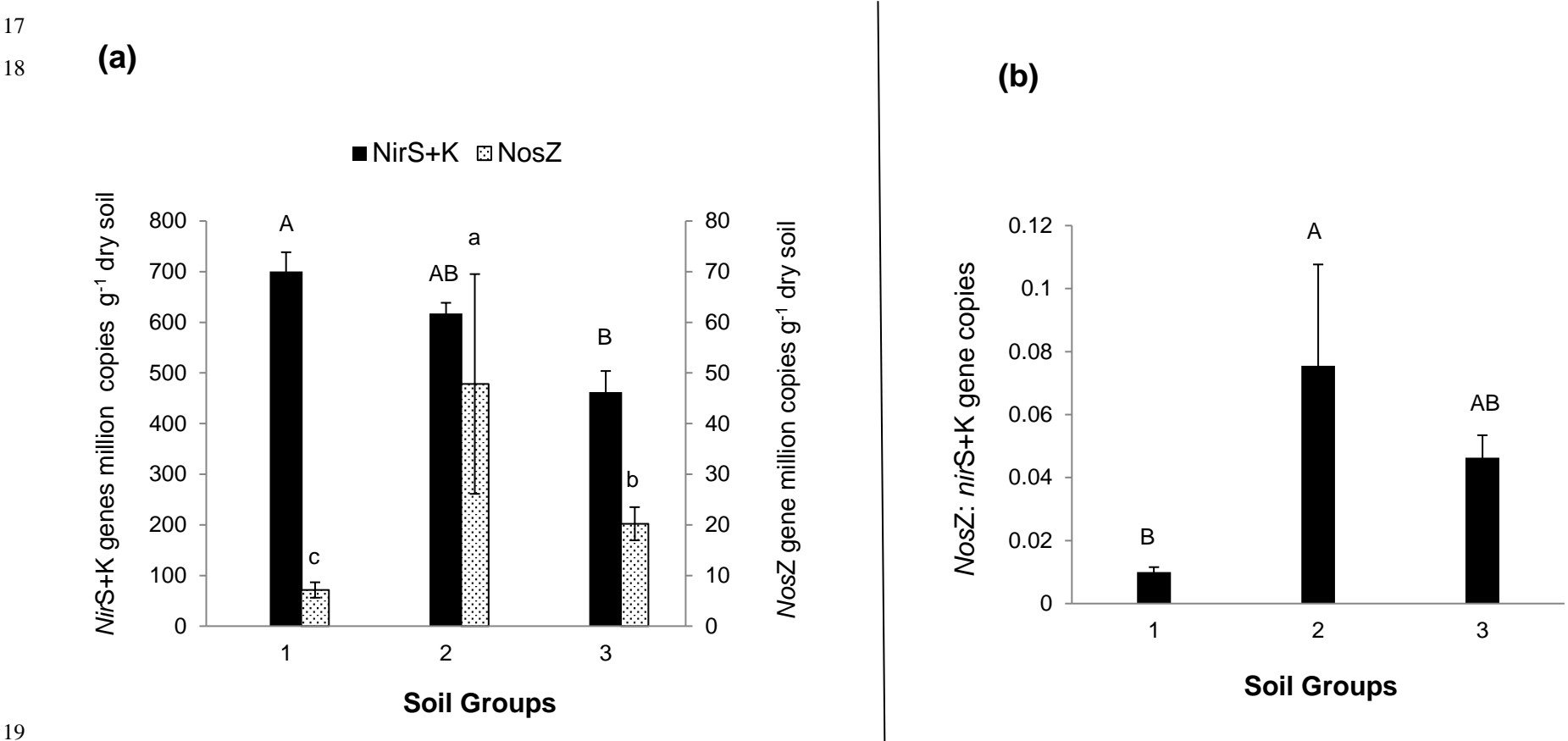

Fig. 4 (a) Denitrifier gene (*nir*S, *nir*K, *nos*Z) copy numbers in different soil groups, error bars denote S.E.M. (b) Denitrifier gene abundance ratio
(*nos*Z : *nir*S, *nir*K) in different soils groups. Mean values are reported ± 1standard error of the mean. Columns with the same letters are not
significantly different. Letter values with same case or font denote one test (one test for each of the group)

**Table 1: Description of soils**

| Soil | Location of the dairy farm | Geographical Location | Soil Abbreviation | Soil Classification | Mineralogy Class |
|---|---|---|---|---|---|
| Te Kowhai Silt Loam | AgResearch Ruakura, Waikato | 37°44'57.55"S 175°10'27.06"E | TeK | Typic Orthic Gley | GlassyVolcanic , Kaolinitic |
| Otorohanga  Silt Loam | Tokanui, Waikato | 38°11'19.70"S 175°12'35.67"E | OH | Typic Orthic | Allophanic |
| Horotiu  Silt Loam | AgResearch Ruakura, Waikato | 37°46'30.80"S 175°18'23.27"E | HR | Typic Orthic Allophanic | Allophanic |
| Tokomaru  Silt Loam | Massey University, Palmerston North | 40°22'58.50"S 175°36'31.01"E | TM | Argillic-fragic Perch-gley Pallic | Vermiculitic |
| Manawatu  Fine Sandy Loam | Longburn, Palmerston North | 40°22'56.99"S 175°32'24.49"E | MW | Weathered fluvial recent | Illitic |
| Manawatu   Fine  Sandy  Loam (Effluent irrigated) | Longburn, Palmerston North | 40°22'58.26"S 175°32'21.65"E | MWEI | Weathered fluvial recent | Illitic |
| Paparua  Silt Loam (Springston) | Springston, Christchurch | 43°38'15.97"S 172°28'13.81"E | PS | Weathered Orthic recent | Illitic |
| Paparua  Silt Loam (Lincoln) | Lincoln, Christchurch | 43°38'43.91"S 172°25'21.86"E | PL | Weathered Orthic recent | Illitic |
| Lismore  Stony Silt Loam | Ashburton, Canterbury | 43°53'17.44"S 171°38'28.43"E | LM | Pallic Orthic Brown | Vermiculitic |
| Mayfield  Deep Silt Loam | Methven, Canterbury | 43°38'30.12"S 171°43'47.28"E | MF | No data | No data |

Adapted from Morales *et al*. 2015