# Peer review of "Soil properties impacting denitrifier community size, structure, and activity in New Zealand dairy-grazed pasture."

_Biogeosciences, 2016_

## Referee Comment (RC1) · Anonymous Referee #1 · 15 Nov 2016

**Comments**

Scientific significance: Does the manuscript represent a substantial contribution to scientific progress within the scope of Biogeosciences (substantial new concepts, ideas, methods, or data)?

The paper content falls within the scope of BG. The objective was to gain insight into relationship between denitrifier community size, structure and activity. This was performed by analyzing genes: nirS, nirK and nosZ. Also denitrifier enzyme activity was analysed. 10 soils each sampled at 6 locations with 25 samples at two depths respectively, and pooled. All analysis was performed later at the laboratory. The study is motivated by N2O emissions, since a potent greenhouse gas, and that complete

denitrification to N2 is better. The authors motivate the study by 'denitrifier community structure is not always strongly correlated to soil or environmental parameters (Dandie et al., 2011;Enwall et al., 15 2010;Philippot et al., 2009) indicating that our understanding of the factors controlling the diversity and function of denitrifying communities is still inadequate.' In contrast Graham et al. (2016 Frontiers in Microbiology) concludes environmental variables are the strongest predictors of process rates, however that microbial data was the next important explanation factors. So what is the hen and the egg? Many new molecular methods have been developed over the last decennia, opening possibilities to study the microbial life in soils. The impression is that the availability of a method designed this study. Results and conclusions are vague.

Scientific quality: Are the scientific approach and applied methods valid? Are the results discussed in an appropriate and balanced way (consideration of related work, including appropriate references)?

The authors are familiar with molecular and microbial genetic and process studies, which were applied here. However one can ask what can the denitrifier community structure tell on the N2O emission size? A DEA assey gives a hint in combination with nosZ genes. But contrasting results were found, where soil of group had low DEA and low nosZ (Fig 4), so what to expect? And soil group 2 high in nosZ where DEA was the highest, does that hint low N2O in spite of high process rate? It is not possible to guess that N2O may be emitted from a soil. This is not discussed in the paper. However N2O emission size was not the main aim of the study, but the study was motivated by it. The motivation of the study is vague (see above), and the objective told in the abstract 'to gain insight to relationships between . . .structure and activity'. What was the insight gained? Ten soils were compared, but one soil (n=1?) is treated as a group of soils (group 2), however many samples at one site. This could be questioned? References to papers describing methods are not appropriate, since the methods are not found there. The Discussion section resembles a Result section however there are references after each paragraph.

Presentation quality: Are the scientific results and conclusions presented in a clear, concise, and well-structured way (number and quality of figures/tables, appropriate use of English language)?

The authors could have better worked the text through. Sometimes the text is difficult to follow. The overall structure is OK, however the content of the discussion could couple more to other work.

Specific comments

P2 L34 This hypothesis is not very visible through the paper. Management practices altering environment conditions at the different soils could not be found.

P3 L6 'Population therefore' something lacking, difficult to read. L17-20 This section describing soil sampling is messy, difficult to read, some things are lacking like only one soil depth here but two depths later on. L23 Standard protocols refers to Morales et al. (2015), but I could not find these methods referred to in this reference. L28 Refers to Morales also for DEA, not in that paper. I have tosay Ihave not checked all references given in the manuscript.

P4 L25 Why was the 10 soils investigated described so sparsely?

P5 L32 Two soils (n=2) compose one group. Enough?

P6 L2 More so for group 2 consisting only one soil.

P7 L12 two orders of magnitude? Only one as I can see.

Many vague and not very clear statements and conclusions, based on one or two soils follows.

---

## Referee Comment (RC2) · Anonymous Referee #2 · 23 Nov 2016

1) Scientific significance: Does the manuscript represent a substantial contribution to scientific progress within the scope of Biogeosciences (substantial new concepts, ideas, methods, or data)?

The manuscript is aiming at unraveling the relationships between denitrifier community structure and environmental parameters in pasture soils. It is well within the focus of the journal. The methods used are solid but not cutting edge and suited to answer some of the questions. However, the experimental design is not perfect for the big aim of understanding the connections between nitrous oxide emissions, denitrifier community structure composition and soil type and land management.

2) Scientific quality: Are the scientific approach and applied methods valid? Are the
results discussed in an appropriate and balanced way (consideration of related work, including appropriate references)?

In principal I think the study has great potential but in present form suffers a little from too many variables between the different soils and not enough samples/replicates of similar soils to resolve their influences.

I further have a slight problem with the determination of copy numbers for functional genes and using these numbers as 'abundances' of the organisms. The denitrifiers could be the same percentage of the total population in all soils and it would make sense to at least also determine the copy numbers of the bacterial 16S rRNA gene with a general primer set. Then there are still issues with gene copy number per genome, functional gene/16S rRNA gene ratio in a genome and such left, which would be harder to account for.

From an organismic point of view it has to be considered that the nirS/K and nosZ genes are not distributed completely independent. They are linked in organisms that can perform the full denitrification pathway. Therefore it is quite surprising that the NMS analysis of nosZ (Fig. 3c) doesn't show any clustering while nirS/K did. Would it be possible to identify the T-RFs of nirS/K that have similar distribution patterns over the samples than those from nosZ? That way only subsets of T-RFs could be analyzed in order to determine how the soil parameters influence their presence/abundance.

The discussion is a bit lackluster and is missing a part in which the results are discussed in the frame of the bigger question, nitrous oxide emissions. Especially as the results of the study seem to suggest that all the soil parameters collected do not explain the distribution and abundance of the nosZ gene over the different soils. How does this fit with the question? I would have expected a more thorough discussion of this, also the potential pitfalls of the methods used that could have influenced this result (primer bias, etc.).

3) Presentation quality: Are the scientific results and conclusions presented in a clear,

concise, and well-structured way (number and quality of figures/tables, appropriate use of English language)?

The quality of the presentation is lacking a little with sentences that sometimes need re-reading before they make sense. Minor grammar mistakes here and there can be found too as well as layout issues.

The figures are not always as informative as they could be.

Figure 1 doesn't resolve the differences between the sites closely located next to each other well. It gives a general impression where the sites are located but why not move it to SOM and then add three zoomed in insert maps that resolve the three local areas where the samples were taken better?

Figure 2 is really busy, especially with the legend for each dot. As the color code already defines which sampling site they are from, why not just put the numbers for the replicates on? And I don't think it adds anything to know which exact replicates are closer together as it is not mentioned elsewhere in the manuscript. So it might be an idea to leave the annotations in the figure off altogether and just rely on the color code explained in the legend. Further, the circles defining the clusters should not cross the borders of the ordination.

Figure 3 is again pretty busy and would need some cleaning up. It would also make sense to stick to the same symbols/colors as in Fig. 2. Fig. 3 b is pretty meaningless as the majority of samples can't be resolved in the presented ordination. Here the question is if an outlier analysis could be used to remove the data points at the edges of the ordination. If not, I would suggest to at least show an ordination with only the data points that cluster tightly together in the SOM to resolve potential trends in this subset of samples that is not affected by the 'outliers'.

The data presented in table 2 would also make a nice figure, maybe even in combination with Fig. 4.

Specific comments

Multiple pages: gene names are normally all italicized, also e.g. the 'K' from 'nirK'.

p 3, l 16: Sampling was conducted between August and December. Where there any kind of controls to test for seasonality effects?

p 3, l 18: Were the 25 soil cores per replicate homogenized and mixed during the process of sieving?

p 3, l 18: Where all samples besides the ones for molecular data stored at 4 °C? If some of the analyses were done 6 months later I would be worried about changes in the soils as microbial activity will continue, although much slower.

p 6, l 8 ff/table 2/figure 3: The number of T-RFs used for the NMS analysis seems to be quite low and in the case of nirK also pretty different between the samples. This could result in problems with the ordination that is hard to evaluate. It would be nice to report stress values and also provide the data matrices used for the NMS analysis in the SOM so the reader can evaluate them.

p 8, l 14: Wouldn't it have been possible to avoid uneven grazing and excretal deposition by fencing off an area a couple of weeks prior to sampling? Or at least try to avoid these spots by a careful screening of the area to find representative spots?

p 9, l 10 ff: I am not sure why the authors are so surprised by this. The sampling procedure (25 cores combined) should diminish the signals from different microniches and create an integrated signal.

p 10, l 21: 'saturated': I assume with water?

p 10, l 24 ff: If the adsorption of copper is the reason that there is less nitrous oxide reduction, then why are there active nirKs, which also have copper as co-factor? There must be another explanation for this observation or could a reduction in the copy numbers of nirK be observed in these soils as well?

---

## Referee Comment (RC3) · Anonymous Referee #3 · 24 Nov 2016

Summary:

They sampled soils from 10 different geographical locations in New Zealand. They did an ordination of soil characteristics and found that the 10 sample locations could be grouped into 3 groups based on soil characteristics. These groupings were used in the further analysis of T-RFLP, qPCR and DEA data.

General comments:

The study attempts to find how various pasture management (soil water, carbon and fertility) will affect the denitrifier community, which increase our knowledge on denitrification in different soil types, and maybe improve our ability to promote complete

denitrification and avoid N2O emission. This is a relevant question within the scope of BG. They find that fertile soil with high microbial biomass promote complete denitrification, whereas allophanic saturated soil is a source of N2O production

I found it hard to get a good overview of the results and discussion, maybe because of poor flow and clarity in writing. I agree with RC1 that the discussion resembles a result section. In general every section sums up observations and have some explanation with a reference. I don't think it reaches a high enough level of discussion. I'm also not confident that the data is strong enough to answer the question sufficiently. qPCR on RNA would be more reliable. To my knowledge the nir genes are very ubiquitous and not necessarily experessed.

Both title and abstract are descriptive and clear, reflecting the study well.

Specific comments:

The whole introduction argumentation for this study (P2, L11 – P3, L2) makes a good background, but somehow it's a bit vague. The idea of the study is very good and this framework can make it more visual with clearer and stronger formulations.

P3, L22-23 I would mention which physicochemical characteristics were used in this study here, otherwise you only see it when reading the statistical analysis.

Regarding methods for physicochemical characteristics, DEA and qPCR, they refer to Morales et al. (2015). This seems to be another study of the very same soil sampling, and this manuscript is reusing data from Morales et al. (2015), right? It should appear more clearly that this study is an extension of Morales et al. (2015) with reuse of data. It would also seem natural to refer more to the earlier study since it's the same topic. There should be references to this in the introduction and/or discussion, not only for methods description.

P10, L25-29 Suddenly in the end of the conclusion this new stuff about allophanic soils comes up, this should have been included earlier on. The conclusion should instead

round and wrap up. New stuff should not be introduced like this.

Technical corrections:

Inconsistent use of water content terms and abbreviations: "Moisture"/"soil water"/"soil water content"/"SWC" and also "% SWC at field capacity"/"% FC SWC"/"high moisture at FC". Also "Field fresh" (P3, L20) and "field-moist" (P3, L22). This was all quite confusing to me.

Figure 2 have too many abbreviations in caption, the figure itself should be more descriptive.

In caption for Figure 4, SEM should first be defined and then used. not the other way around.

P1, L3 There should not be a dot in the end of the manuscript title. This also occurs in the titles in the references.

P2, L34 With enhanced structure, do you then mean diversity?

P3, L19 "2 depths" not "2 depth". I can't find which depths you chose (mm/cm?), should be stated in the methods.

P4, L7-8 "2.5 ul of 10xPCR buffer (1 mM MgCl2), 0.5 mM MgCl2". Final concentrations in reaction mix should be stated, this looks weird to me.

P4, L24 I would specify that the qPCR was performed on DNA

P5, L19 Isn't the right abbreviation NMDS? Not NMS

---

## Author Comment (AC1) · 22 Mar 2017

Response to reviewers' comments:

Soil properties impacting denitrifier community size, structure and activity in New Zealand dairy-grazed pasture

Neha Jha, Surinder Saggar, Donna Giltrap, Russ Tillman, and Julie Deslippe

In this document we provide a comprehensive description of how we have responded to all the changes suggested by the associate editor.

Anonymous Referee #1 Comments

Scientific significance: Does the manuscript represent a substantial contribution to scientific progress within the scope of Biogeosciences (substantial new concepts, ideas, methods, or data)?

The paper content falls within the scope of BG. The objective was to gain insight into relationship between denitrifier community size, structure and activity. This was performed by analyzing genes: nirS, nirK and nosZ. Also denitrifier enzyme activity was analysed. 10 soils each sampled at 6 locations with 25 samples at two depths respectively, and pooled. All analysis was performed later at the laboratory. The study is motivated by N2O emissions, since a potent greenhouse gas, and that complete denitrification to N2 is better. The authors motivate the study by 'denitrifier community structure is not always strongly correlated to soil or environmental parameters (Dandie et al., 2011;Enwall et al., 15 2010;Philippot et al., 2009) indicating that our understanding of the factors controlling the diversity and function of denitrifying communities is still inadequate.' In contrast Graham et al. (2016 Frontiers in Microbiology) concludes environmental variables are the strongest predictors of process rates, however that microbial data was the next important explanation factors. So what is the hen and the egg?

Author's Response - Thank you for drawing our attention to this important synthesis. Graham et al 2016 address the question "When do we need to accurately predict microbial community structure to accurately predict function?" In this re-analysis of 82 existing datasets of bacterial community structure and a variety of ecosystem processes (both C and N cycling) the authors show that microbial community metrics had low power to explain ecosystem process rates but they improved models based on environmental variables alone by on average 8%, which while significant is admittedly not stellar. In particular, they found that models based on all predictor sets (environmental variables only, microbial parameters only, or environmental + microbial parameters) had very low power to explain denitrification rates but that community diversity metrics added more explanatory power for denitrification rates than for any other process

(which partly justifies our approach). The aim of our study was to achieve a better understanding of the relationships between the structure, abundance, and activity of denitrifiers over a range of dairy-pasture soils. As justification of this aim we suggest that this 'may enhance our ability to promote complete denitrification in order to reduce N2O emissions from pastoral agriculture'. Given the results of Graham et al 2016 we concede that this now seems overly optimistic and we have revised the Introduction to reflect this, however, we point out that the former study did not directly analyse N2O:N2 ratios during denitrification. We have also made a large number of revisions to refocus the manuscript on our central question which is 'if the size and activity of bacterial denitrifying communities can be predicted on the basis of soil physicochemical characteristics'. We feel that this is clearly a separate question than that addressed by Graham et al. but one that could shed additional light on the environmental contexts wherein microbial community structure and diversity can inform ecosystem function.

Many new molecular methods have been developed over the last decennia, opening possibilities to study the microbial life in soils. The impression is that the availability of a method designed this study. Results and conclusions are vague.

Author's Response - This is unfortunate and points clearly to the need for a thorough revision of our manuscript in order to better frame its goal. In response to this comment we have completely revised the results and discussion.

Scientific quality: Are the scientific approach and applied methods valid? Are the results discussed in an appropriate and balanced way (consideration of related work, including appropriate references)?

The authors are familiar with molecular and microbial genetic and process studies, which were applied here.However one can ask what can the denitrifier community structure tell on the N2O emission size?

Author's Response - Here we present qPCR data for the number of gene copies for the functional genes nirS, nirK and nosZ, as well as for the ratio on nirS+K:nosZ. The ratio

of nir:noz genes has been interpreted previously as an index of the potential for complete denitrificiation (Phillipot et al 2011, Braker er al 2012, Jones et al 2014). Generally, it is expected that soils with low nir:nos ratios are more likely to emit proportionally smaller N as N2O. We have now clarified this in the methods section.

A DEA assay gives a hint in combination with nosZ genes. But contrasting results were found, where soil of group had low DEA and low nosZ (Fig 4), so what to expect? And soil group 2 high in nosZ where DEA was the highest, does that hint low N2O in spite of high process rate?

Author's Response - Group 2 soils (based on soil physiochemical characteristics) varied widely with regard to both DEA and the number of nosZ gene copies (fig 2 and fig 4) but within a soil these parameters largely agreed. This agreement drove the significant positive correlation among DEA and nosZ copy numbers which we report in supplementary table S3. Both high DEA and higher nosZ gene copy might indicate low N2O despite high denitrification rate under most favourable condition in these soils. The revised discussion is substantially clearer on this point.

It is not possible to guess that N2O may be emitted from a soil. This is not discussed in the paper. However N2O emission size was not the main aim of the study, but the study was motivated by it. The motivation of the study is vague (see above), and the objective told in the abstract 'to gain insight to relationships between . . .structure and activity'.

Author's Response - As above, we have rewritten the introduction section to deemphasise a direct link between denitrifier community size/structure and N2O emissions from soils.

What was the insight gained? Ten soils were compared, but one soil (n=1?) is treated as a group of soils (group 2), however many samples at one site. This could be questioned?

Author's Response - The soils grouped into 3 distinct clusters based on their physic-ochemical characteristics. This is a result, not an aspect of our sampling design. We then ask whether microbial community diversity, structure and size varied according to these same major gradients in physicochemical characteristics. We find that they do not, but rather responded primarily to soil water content and Olsen P. This is much more clearly communicated in the revised manuscript.

References to papers describing methods are not appropriate, since the methods are not found there.

Author's Response - Thank you, we have replaced the erroneous reference with the correct one.

The Discussion section resembles a Result section however there are references after each paragraph. The authors could have better worked the text through. Sometimes the text is difficult to follow. The overall structure is OK, however the content of the discussion could couple more to other work.

Author's Response - As above, we have thoroughly revised the results and discussion sections.

Specific comments P2 L34 This hypothesis is not very visible through the paper. Management practices altering environment conditions at the different soils could not be found.

Given the centrality of soil water content in driving bacterial denitrifier community metrics in our study we have modified the discussion section to include a more thorough discussion of the ways in which pasture management can influence soil water content.

P3 L6 'Population therefore' something lacking, difficult to read.

Revised.

L17-20 This section describing soil sampling is messy, difficult to read, some things are

lacking like only one soil depth here but two depths later on.

Additional information has been included to clarify the soil sampling.

L23 Standard protocols refers to Morales et al. (2015), but I could not find these methods referred to in this reference. L28 Refers to Morales also for DEA, not in that paper. I have to say I have not checked all references given in the manuscript.

As above, we have replaced this erroneous reference with the correct one.

P4 L25 Why was the 10 soils investigated described so sparsely?

Detailed description of the 10 soils investigated has been provided in the supplementary section.

P5 L32 Two soils (n=2) compose one group. Enough? P6 L2 More so for group 2 consisting only one soil.

As above, the soils grouped into 3 distinct clusters based on their physicochemical characteristics. This is a result, not an aspect of our sampling design.

P7 L12 two orders of magnitude? Only one as I can see.

Thank you, this was a typo that has been corrected.

---

## Author Comment (AC3) · 3 Apr 2017

3 April, 2017

Re: Final comments to the Associate Editor of Biogeosciences

Dear Sir/Madam

Thank you very much for the opportunity to submit our manuscript 'Soil properties impacting denitrifier community size, structure and activity in New Zealand dairy-grazed pasture' (bg-2016-390) to Biogeosciences. We are very grateful for the insightful comments you and the three reviewers have made, which made it clear to us that the manuscript required a major overhaul.

[Figure]

We also wish to express our gratitude to you and your editorial team for the flexibility you have shown us with regard to the timeline of this review. An unusual combinations of events, which include two newborn babies and a heart attract, affected our team and so our energies were diverted from this manuscript longer than we would have wished. Now however, we are pleased to report that we have very carefully considered and responded to each of the reviewer's comments. These efforts have resulted in a vastly improved manuscript, including changes to all figures, a new figure, an overhaul of the introduction, methods and results sections and a complete rewrite of the discussion and conclusions. While the main results of the manuscript are unchanged, they are much more clearly communicated, and we feel confident that you and the reviewers will be satisfied with this progress.

Here we characterise the size, structure and diversity of nirS, nirK and nosZ genes in soils that varied widely in physicochemical characteristics to address the question of whether different denitrifier communities develop under these varied soil conditions, and if so, whether they are associated with different denitrification activities and likely to generate different N2O emissions. Overall, we found a strong correlation between MBC and DEA and that moderately high to highly fertile soils supported the largest populations of denitrifiers. Given that the more fertile soils were also likely to harbour significant populations of nitrifiers MBC may be an important coarse-scale indicator of total potential N2O emissions from such soils. However, our results for allophanic soils suggest that even relatively low rates of denitrification may lead to significant N2O emissions given their relatively low nos:nir. Consequently, we conclude that management strategies to limit N2O emissions through denitrification are likely to be most important for dairy farms on fertile or allophanic soils during wetter periods. Finally, our data suggest that new techniques that would selectively target nirS denitrifiers may be the most effective for limiting N2O emissions through denitrification across a wide range of soil types.

We eagerly look forward to the opportunity to submit our revised manuscript to Biogeo-

sciences.

With very best wishes,

Dr. Neha Jha (On behalf of: Surinder Saggar, Donna Giltrap, Russ Tillman, and Julie Deslippe)

Response to reviewers' comments:

Soil properties impacting denitrifier community size, structure and activity in New Zealand dairy-grazed pasture.

Neha Jha, Surinder Saggar, Donna Giltrap, Russ Tillman, and Julie Deslippe

In this document we provide a comprehensive description of how we have responded to all the changes suggested by the associate editor.

Anonymous Referee #1 Comments

**Scientific significance: Does the manuscript represent a substantial contribution to scientific progress within the scope of Biogeosciences (substantial new concepts, ideas, methods, or data)?**

**The paper content falls within the scope of BG. The objective was to gain insight into relationship between denitrifier community size, structure and activity. This was performed by analyzing genes: nirS, nirK and nosZ. Also denitrifier enzyme activity was analysed. 10 soils each sampled at 6 locations with 25 samples at two depths respectively, and pooled. All analysis was performed later at the laboratory.**

**The study is motivated by N2O emissions, since a potent greenhouse gas, and that complete denitrification to N2 is better. The authors motivate the study by 'denitrifier community structure is not always strongly correlated to soil or environmental parameters (Dandie et al., 2011;Enwall et al., 15 2010;Philippot et al., 2009) indicating that our understanding of the factors controlling the diversity and function of denitrifying communities is still inadequate.' In contrast Graham et al. (2016 Frontiers in Microbiology)**

concludes environmental variables are the strongest predictors of process rates, how-
ever that microbial data was the next important explanation factors. So what is the hen
and the egg?

Author's Response: Thank you for drawing our attention to this important synthesis.
Graham et al. 2016 address the question "When do we need to accurately predict
microbial community structure to accurately predict function?" In this re-analysis of
82 existing datasets of bacterial community structure and a variety of ecosystem pro-
cesses (both C and N cycling) the authors show that microbial community metrics had
low power to explain ecosystem process rates but they improved models based on en-
vironmental variables alone by on average 8%, which while significant is admittedly not
stellar.

In particular, they found that models based on all predictor sets (environmental vari-
ables only, microbial parameters only, or environmental + microbial parameters) had
very low power to explain denitrification rates but that community diversity metrics
added more explanatory power for denitrification rates than for any other process
(which partly justifies our approach). The aim of our study was to achieve a better
understanding of the relationships between the structure, abundance, and activity of
denitrifiers over a range of dairy-pasture soils. As justification of this aim we suggest
that this 'may enhance our ability to promote complete denitrification in order to reduce
N2O emissions from pastoral agriculture'.

Given the results of Graham et al. 2016 we concede that this now seems overly opti-
mistic and we have revised the introduction to reflect this, however, we point out that
the former study did not directly analyse N2O:N2 ratios during denitrification. We have
also made a large number of revisions to refocus the manuscript on our central ques-
tion which is 'if the size and activity of bacterial denitrifying communities can be pre-
dicted on the basis of soil physicochemical characteristics'. We feel that this is clearly
a separate question than that addressed by Graham et al. but one that could shed
additional light on the environmental contexts wherein microbial community structure

and diversity can inform ecosystem function.

**Many new molecular methods have been developed over the last decennia, opening possibilities to study the microbial life in soils. The impression is that the availability of a method designed this study. Results and conclusions are vague.**

Author's Response: This is unfortunate and points clearly to the need for a thorough revision of our manuscript in order to better frame its goal. In response to this comment we have completely revised the results and discussion.

**Scientific quality: Are the scientific approach and applied methods valid? Are the results discussed in an appropriate and balanced way (consideration of related work, including appropriate references)?**

**The authors are familiar with molecular and microbial genetic and process studies, which were applied here. However one can ask what can the denitrifier community structure tell on the N2O emission size?**

Author's Response: Here we present qPCR data for the number of gene copies for the functional genes nirS, nirK and nosZ, as well as for the ratio of nos: nir. The ratio of these genes has been interpreted previously as an index of the potential for complete denitrification (Phillipot et al. 2011, Braker et al. 2012, Jones et al. 2014). Generally, it is expected that soils with high nos: nir ratios are more likely to emit proportionally smaller N as N2O. We have now clarified this in the methods section.

**A DEA assay gives a hint in combination with nosZ genes. But contrasting results were found, where soil of group had low DEA and low nosZ (Fig 4), so what to expect? And soil group 2 high in nosZ where DEA was the highest, does that hint low N2O in spite of high process rate?**

Author's Response: Group 2 soils (based on soil physicochemical characteristics) varied widely with regard to both denitrification enzyme activity (DEA) and the number of nosZ gene copies (fig 2 and fig 4) but within a soil these parameters largely agreed.

This agreement drove the significant positive correlation among DEA and nosZ copy numbers which we report in supplementary table S3. Both high DEA and higher nosZ gene copy might indicate low N2O despite high denitrification rate under most favourable condition in these soils. The revised discussion is substantially clearer on this point.

**It is not possible to guess that N2O may be emitted from a soil. This is not discussed in the paper. However N2O emission size was not the main aim of the study, but the study was motivated by it. The motivation of the study is vague (see above), and the objective told in the abstract 'to gain insight to relationships between structure and activity'.**

Author's Response: As above, we have rewritten the introduction section to deemphasise a direct link between denitrifier community size/structure and N2O emissions from soils.

**What was the insight gained? Ten soils were compared, but one soil (n=1?) is treated as a group of soils (group 2), however many samples at one site. This could be questioned?**

Author's Response: The soils grouped into 3 distinct clusters based on their physicochemical characteristics. This is a result, not an aspect of our sampling design. We then ask whether microbial community diversity, structure and size varied according to these same major gradients in physicochemical characteristics. We find that they do not, but rather responded primarily to soil water content and Olsen P. This is much more clearly communicated in the revised manuscript.

**References to papers describing methods are not appropriate, since the methods are not found there.**

Author's Response: Thank you, we have replaced the erroneous reference with the correct one.

\# The Discussion section resembles a Result section however there are references after each paragraph.

Author's Response: As above, we have thoroughly revised the results and discussion sections.

\# Presentation quality: Are the scientific results and conclusions presented in a clear, concise, and well-structured way (number and quality of figures/tables, appropriate use of English language)?

\# The authors could have better worked the text through. Sometimes the text is difficult to follow. The overall structure is OK, however the content of the discussion could couple more to other work.

Author's Response: Thank you, we have thoroughly revised the results and discussion sections.

\# Specific comments

P2 L34 This hypothesis is not very visible through the paper. Management practices altering environment conditions at the different soils could not be found.

Author's Response: Given the centrality of soil water content in driving bacterial denitrifier community metrics in our study we have modified the discussion section to include a more thorough discussion of the ways in which pasture management can influence soil water content.

P3 L6 'Population therefore' something lacking, difficult to read.

Author's Response: Revised.

L17-20 This section describing soil sampling is messy, difficult to read, some things are lacking like only one soil depth here but two depths later on.

Author's Response: Additional information has been included to clarify the soil sampling.

L23 Standard protocols refers to Morales et al. (2015), but I could not find these methods referred to in this reference. L28 Refers to Morales also for DEA, not in that paper. I have to say I have not checked all references given in the manuscript.

Author's Response: As above, we have replaced this erroneous reference with the correct one.

P4 L25 Why was the 10 soils investigated described so sparsely?

Author's Response: Detailed description of the 10 soils investigated has been provided in the supplementary section.

P5 L32 Two soils (n=2) compose one group. Enough? P6 L2 More so for group 2 consisting only one soil.

Author's Response: As above, the soils grouped into 3 distinct clusters based on their physicochemical characteristics. This is a result, not an aspect of our sampling design.

P7 L12 two orders of magnitude? Only one as I can see.

Author's Response: Thank you, this was a typo that has been corrected.

Many vague and not very clear statements and conclusions, based on one or two soils follows.

Author's Response: We have thoroughly revised the manuscript to avoid any vague or unclear statement.

Anonymous Referee #2

 1)

**Scientific significance: Does the manuscript represent a substantial contribution to scientific progress within the scope of Biogeosciences (substantial new concepts, ideas, methods, or data)?**

[Figure]

\# The manuscript is aiming at unravelling the relationships between denitrifier community structure and environmental parameters in pasture soils. It is well within the focus of the journal. The methods used are solid but not cutting edge and suited to answer some of the questions. However, the experimental design is not perfect for the big aim of understanding the connections between nitrous oxide emissions, denitrifier community structure composition and soil type and land management.

Author's Response: As in our responses to R1 above, we concede that our aim of understanding the link between the structure, abundance, and activity of denitrifiers based on soil physicochemical characteristics may not directly 'enhance our ability to promote complete denitrification in order to reduce N2O emissions from pastoral agriculture' and we have now revised the introduction to reflect this.

\# Scientific quality: Are the scientific approach and applied methods valid? Are the results discussed in an appropriate and balanced way (consideration of related work, including appropriate references)?

\# In principal I think the study has great potential but in present form suffers a little from too many variables between the different soils and not enough samples/replicates of similar soils to resolve their influences.

Author's Response: We present n=6 for all soil physicochemical datasets and n=3 for molecular microbial datasets. However molecular work was based on 6 separate DNA extractions followed by pooling 2 extractions/PCR amplification in attempt to better represent potential spatial variability among replicates.

\# I further have a slight problem with the determination of copy numbers for functional genes and using these numbers as 'abundances' of the organisms. The denitrifiers could be the same percentage of the total population in all soils and it would make sense to at least also determine the copy numbers of the bacterial 16S rRNA gene with a general primer set. Then there are still issues with gene copy number per genome, functional gene/16S rRNA gene ratio in a genome and such left, which would be harder

to account for.

Author's Response: Yes, agreed. This problem is inherent in many qPCR studies of functional genes. We have revised the methods and results sections to reflect this limitation of our approach. In particular, we have moved figure 4a to the supplementary data so that our results and discussion focus on the nos: nir ratio only. Because these genes do not always (but can) co-occur within an organism their ratio may better reflect cell numbers of complete: incomplete denitrifiers. Of course, this assumes similar PCR bias among the different primer sets, but that assumption applies equally to amplification of a "housekeeper gene" like 16S rRNA or rpoB.

**From an organismic point of view it has to be considered that the nirS/K and nosZ genes are not distributed completely independent. They are linked in organisms that can perform the full denitrification pathway. Therefore it is quite surprising that the NMS analysis of nosZ (Fig. 3c) doesn't show any clustering while nirS/K did. Would it be possible to identify the T-RFs of nirS/K that have similar distribution patterns over the samples than those from nosZ? That way only subsets of T-RFs could be analyzed in order to determine how the soil parameters influence their presence/abundance.**

Author's Response: This is an interesting suggestion. It would certainly shed light on the how complete denitrifiers respond to varied soil conditions. However, this is really a separate question from the one we pose here because complete denitrifiers are typically only a small subset ($\sim$ 0.5%; Deslippe et al. 2014) of all denitrifiers in New Zealand pasture soils. Should we follow this suggestion, we would miss incomplete denitrifiers, which are equally likely to be affected by the soil physicochemical characteristics we study here, and they are especially of concern for GHG emissions.

**The discussion is a bit lackluster and is missing a part in which the results are discussed in the frame of the bigger question, nitrous oxide emissions. Especially as the results of the study seem to suggest that all the soil parameters collected do not explain the distribution and abundance of the nosZ gene over the different soils. How does this**

fit with the question? I would have expected a more thorough discussion of this, also the potential pitfalls of the methods used that could have influenced this result (primer bias, etc.).

Author's Response: Yes agreed. We have thoroughly revised the discussion section and we now more fully address reasons that the distribution and abundance of nosZ genes respond primarily to SWC and Olsen P in our study.

**Presentation quality: Are the scientific results and conclusions presented in a clear, concise, and well-structured way (number and quality of figures/tables, appropriate use of English language)?**

**The quality of the presentation is lacking a little with sentences that sometimes need re-reading before they make sense. Minor grammar mistakes here and there can be found too as well as layout issues.**

Author's Response: We have given the manuscript a general overhaul and respond to specific issues in detail below.

**The figures are not always as informative as they could be.**

**Figure 1 doesn't resolve the differences between the sites closely located next to each other well. It gives a general impression where the sites are located but why not move it to SOM and then add three zoomed in insert maps that resolve the three local areas where the samples were taken better?**

Author's Response: Thanks we have revised this figure.

**Figure 2 is really busy, especially with the legend for each dot. As the color code already defines which sampling site they are from, why not just put the numbers for the replicates on? And I don't think it adds anything to know which exact replicates are closer together as it is not mentioned elsewhere in the manuscript. So it might be an idea to leave the annotations in the figure off altogether and just rely on the color code explained in the legend. Further, the circles defining the clusters should not cross the**

borders of the ordination.

Author's Response: Done. Good suggestion, thanks.

**Figure 3 is again pretty busy and would need some cleaning up. It would also make sense to stick to the same symbols/colors as in Fig. 2. Fig. 3 b is pretty meaningless as the majority of samples can't be resolved in the presented ordination. Here the question is if an outlier analysis could be used to remove the data points at the edges of the ordination. If not, I would suggest to at least show an ordination with only the data points that cluster tightly together in the SOM to resolve potential trends in this subset of samples that is not affected by the 'outliers'.**

Author's Response: In this version of the manuscript we have recreated figure 3 (now figure 4). It now retains the same symbol colours as in figure 2, but has different symbol shapes to communicate the soil groups (based on the PCA result). We disagree that nirS ordination (formerly fig3b) is meaningless because it illustrates that nirK community structure responded to the same physicochemical characteristics (SWC and Olsen P) as nirS communities did, which is a major point of the manuscript. However we acknowledge that the importance of this result was not sufficiently described in the previous results section nor was it adequately discussed. Consequently, in this version of the manuscript we have corrected those issues as well. While we disagree that outlier analysis is appropriate in this case (removal of HR and PL soils constitutes a 20% data reduction), as requested we have, added an ordination of the nirK data without PL and HR soils to the supplementary materials, which shows that Olsen P and soil water variables remain the primary driver of nirK community structure, even for this reduced dataset. Likewise, we have added this information to the results and discussion.

**The data presented in table 2 would also make a nice figure, maybe even in combination with Fig. 4.**

Author's Response: Agreed. Since the patterns of significance were similar for gene richness, evenness and diversity we chose, (for the sake of simplicity) to make a figure

illustrating only gene richness by soil group. We have moved table 2 to the supplementary section.

**Specific comments**

**Multiple pages: gene names are normally all italicized, also e.g. the 'K' from 'nirK'**

Author's Response: Thanks, we have now thoroughly checked the manuscript for italicized gene name.

p 3, l 16: Sampling was conducted between August and December. Where there any kind of controls to test for seasonality effects?

Author's Response: Our aim was to sample from the range of soil conditions that occur on NZ pasture farms. It was therefore important to sample in both wet and dry seasons. However it was not our intention to characterise the amplitude of seasonal variation within any given soil, and so we did not design controls that would allow us to assess seasonal variation. However, to ensure that our sampling spanned the range of soil moistures that are typical for pasture soils in NZ we sampled the soils that were expected to be wettest (OH and TeK) in winter and the soils that were expected to be driest in summer (PS, LM MF) the other soils were sampled in between these. We have clarified this in the methods section

p 3, l 18: Were the 25 soil cores per replicate homogenized and mixed during the process of sieving?

Author's Response: Yes. Thanks for pointing out that this was unclear. This information has been added to the methods section.

p 3, l 18: Were all samples besides the ones for molecular data stored at 4 âŮęC? If some of the analyses were done 6 months later I would be worried about changes in the soils as microbial activity will continue, although much slower.

Author's Response: The soils used in this study were collected over a nearly 6-month

period. After each soil was sampled it was immediately sieved and pH, nitrate (NO3–) & ammonium (NH4+) –N (mineral-N), total nitrogen (TN), total carbon (TC), Olsen phosphorus (P), microbial biomass carbon (MBC), soluble C, and denitrification enzyme activity (DEA) were measured within 1-2 weeks. Measurements of DEA and MBC were prioritized so that they typically occurred within the first few days after sieving. However, after the first two sets of soils were sampled, a technical problem with our analytical set-up caused delay in measurements of MBC for nearly 3 months. Given that it was not possible to go back to farms and resample all soils (as their physicochemical properties were likely to have changed in this time, we remeasured MBC on the initially sampled and stored soils after 3, 4 and 7 months, we determined that no significant changes in MBC occurred between the time period of 3 and 7 months. We therefore report MBC data for all soils that were stored between 4 and 6 months. We understand that this issue can be confusing to readers so we have clarified this in the methods section, as simply as possible.

p 6, l 8 ff/table 2/figure 3: The number of T-RFs used for the NMS analysis seems to be quite low and in the case of nirK also pretty different between the samples. This could result in problems with the ordination that is hard to evaluate. It would be nice to report stress values and also provide the data matrices used for the NMS analysis in the SOM so the reader can evaluate them.

Author's Response: Thank you for this useful comment. Total T-RF richness was nirS=52, nirK=53, nosZ=47, which is quite typical for T-RFLP studies of functional genes. However, you are correct that the minimum and maximum number of T-RF varied among samples, which could possibly have contributed to instability in the NMDS ordinations we present. Final stress for the three ordinations in fig 3 were as follows: nirS=12.5, nirK=5.5, nosZ=9.4. So this was clearly not a major problem in our datasets. Nevertheless this is a good point and we have added this information to the discussion and SOM sections. We would also point out that the new T-RF richness figure (and specifically the size of the error bars on the histograms), which we have produced in

response to your earlier comment, will also help our readers to evaluate variability in gene richness among samples in our study.

p 8, l 14: Wouldn't it have been possible to avoid uneven grazing and excretal deposition by fencing off an area a couple of weeks prior to sampling? Or at least try to avoid these spots by a careful screening of the area to find representative spots?

Author's Response: All of the pastures sampled in this study were fenced from livestock and none had been grazed within 8 weeks of sampling. Thank you for pointing out this omission; this has now been added to the methods section. As explained in the methods section we also avoided any (old) dung patches when sampling, as bovine gut bacteria could have contaminated the soil sample if we had pushed the soil corer through a dung pile, and so we did not do this.

p 9, l 10 ff: I am not sure why the authors are so surprised by this. The sampling procedure (25 cores combined) should diminish the signals from different microniches and create an integrated signal.

Author's Response: True, but as we say we would then expect nirS and nirK to be negatively correlated overall. No significant negative correlation between nirS and nirK suggests independent environmental or stochastic controls on the size of these populations. This section has been expanded in the revised discussion.

p 10, l 21: 'saturated': I assume with water?

Author's Response: Yes, clarified.

p 10, l 24 ff: If the adsorption of copper is the reason that there is less nitrous oxide reduction, then why are there active nirKs, which also have copper as co-factor? There must be another explanation for this observation or could a reduction in the copy numbers of nirK be observed in these soils as well?

Author's Response: Yes, thanks for pointing out that our argument was confusing. We have revised the conclusions to make the point clearer. We did not intend to suggest

that adsorption of copper is the reason that there is less nitrous oxide reduction in allophanic soils, but rather less nitrite reduction. We agree that because allophanic soils adsorb copper, they are likely to select against nirK denitrifiers. We expected this to reduce the overall number of genes encoding nitrite reductase in group 1 soils, but we didn't observe this (Fig 4). We have revised the discussion section of the manuscript to include this point. The point of interest in the conclusion section is that, the nos:nir gene ratio data we show agrees with previous work by our group showing that allophanic soils emit greater N2O: (N2O + N2O) relative to other soil types.

Anonymous Referee #3

**Summary:**

**They sampled soils from 10 different geographical locations in New Zealand. They did an ordination of soil characteristics and found that the 10 sample locations could be grouped into 3 groups based on soil characteristics. These groupings were used in the further analysis of T-RFLP, qPCR and DEA data.**

**General comments:**

**The study attempts to find how various pasture management (soil water, carbon and fertility) will affect the denitrifier community, which increase our knowledge on denitrification in different soil types, and maybe improve our ability to promote complete denitrification and avoid N2O emission. This is a relevant question within the scope of BG. They find that fertile soil with high microbial biomass promote complete denitrification, whereas allophanic saturated soil is a source of N2O production.**

**I found it hard to get a good overview of the results and discussion, maybe because of poor flow and clarity in writing. I agree with RC1 that the discussion resembles a result section. In general every section sums up observations and have some explanation with a reference. I don't think it reaches a high enough level of discussion. I'm also not**

confident that the data is strong enough to answer the question sufficiently. qPCR on RNA would be more reliable. To my knowledge the nir genes are very ubiquitous and not necessarily expressed.

Author's Response: We agree with R3's assessment that these doubts stem from poor flow of the manuscript and a lack of clarity in writing. These comments align with those of the other reviewers and made it clear to us that the manuscript required a major overhaul. To this end we have revised all parts of our manuscript as detailed elsewhere. Now that this is done we feel that our aim of achieving a better understanding of how soil physicochemical characteristics' affect the size, structure and activity of bacterial denitrifying communities is clear, and we think that R3 would agree that qPCR of RNA would not be an appropriate tool with which to address it.

\# Both title and abstract are descriptive and clear, reflecting the study well.

\# Specific comments:

\# The whole introduction argumentation for this study (P2, L11 – P3, L2) makes a good background, but somehow it's a bit vague. The idea of the study is very good and this framework can make it more visual with clearer and stronger formulations.

Author's Response: To this end we have added to the introduction one sentence, immediately after the statement of aim: "In particular, we asked if the size and activity of bacterial denitrifiers could be predicted on the basis of soil physicochemical characteristics."

P3, L22-23 I would mention which physicochemical characteristics were used in this study here, otherwise you only see it when reading the statistical analysis.

Author's Response: Added, thank you.

Regarding methods for physicochemical characteristics, DEA and qPCR, they refer to Morales et al. (2015). This seems to be another study of the very same soil sampling, and this manuscript is reusing data from Morales et al. (2015), right? It should appear
more clearly that this study is an extension of Morales et al. (2015) with reuse of data. It would also seem natural to refer more to the earlier study since it's the same topic. There should be references to this in the introduction and/or discussion, not only for methods description.

Author's Response: Yes that's right, some of the physicochemical and molecular data presented here also appears in Morales et al. (2015), although the data analysis and objective of both the studies is entirely different. We revised the methods to more clearly convey that point, we also now refer to the Morales et al. (2015) in introduction of our paper.

P10, L25-29 Suddenly in the end of the conclusion this new stuff about allophanic soils comes up, this should have been included earlier on. The conclusion should instead round and wrap up. New stuff should not be introduced like this.

Author's Response: Yes, we agree and include the point about N2O emissions from allophanic soils in the discussion too.

**Technical corrections:**

**Inconsistent use of water content terms and abbreviations: "Moisture"/"soil water"/"soil water content"/"SWC" and also "% SWC at field capacity"/"% FC SWC"/"high moisture at FC". Also "Field fresh" (P3, L20) and "field-moist" (P3, L22). This was all quite confusing to me.**

Author's Response: Thank you, we have revised all parts of the manuscript with an eye for consistency.

**Figure 2 have too many abbreviations in caption, the figure itself should be more descriptive.**

Author's Response: This same comment was made by R2 and so we have changed Fig 2 accordingly.

[Figure]

**In caption for Figure 4, SEM should first be defined and then used. Not the other way around.**

Author's Response: Agreed, done that.

P1, L3 There should not be a dot in the end of the manuscript title. This also occurs in the titles in the references.

Author's Response: We have removed dot from the end of the manuscript title and also from the titles in the references.

P2, L34 With enhanced structure, do you then mean diversity?

Author's Response: This comment has been rephrased for clarity.

P3, L19 "2 depths" not "2 depth". I can't find which depths you chose (mm/cm?), should be stated in the methods.

Author's Response: Yes thanks, we have fixed this and also added the unit of measurement.

P4, L7-8 "2.5 ul of 10xPCR buffer (1 mM MgCl2), 0.5 mM MgCl2". Final concentrations in reaction mix should be stated, this looks weird to me.

Author's Response: Okay, we have rewritten as final molarity. P4, L24 I would specify that the qPCR was performed on DNA

Author's Response: The title of the section "Quantitative polymerase chain reaction (qPCR) of total bacterial and denitrifier genes" makes this point clear.

P5, L19 Isn't the right abbreviation NMDS? Not NMS

Author's Response: Both abbreviations are in common use, with variation stemming from the term used by the particular stats package. PCOrd software refers to NMS ordinations (McCune and Grace, 2002), thus our use of that abbreviation here.

---

## Author Response (AR1)

**Response to reviewers' comments:**

**Soil properties impacting denitrifier community size, structure and activity in New Zealand dairy-grazed pasture.**

Neha Jha, Surinder Saggar, Donna Giltrap, Russ Tillman, and Julie Deslippe

In this document we provide a comprehensive description of how we have responded to all the changes suggested by the associate editor.

**Anonymous Referee #1**

Comments

**Scientific significance: Does the manuscript represent a substantial contribution to scientific progress within the scope of Biogeosciences (substantial new concepts, ideas, methods, or data)?**

The paper content falls within the scope of BG. The objective was to gain insight into relationship between denitrifier community size, structure and activity. This was performed by analyzing genes: nirS, nirK and nosZ. Also denitrifier enzyme activity was analysed. 10 soils each sampled at 6 locations with 25 samples at two depths respectively, and pooled. All analysis was performed later at the laboratory.

The study is motivated by $N_2O$ emissions, since a potent greenhouse gas, and that complete denitrification to $N_2$ is better. The authors motivate the study by 'denitrifier community structure is not always strongly correlated to soil or environmental parameters (Dandie et al., 2011;Enwall et al., 15 2010;Philippot et al., 2009) indicating that our understanding of the factors controlling the diversity and function of denitrifying communities is still inadequate.' In contrast Graham et al. (2016 Frontiers in Microbiology) concludes environmental variables are the strongest predictors of process rates, however that microbial data was the next important explanation factors. So what is the hen and the egg?

Author's Response: Thank you for drawing our attention to this important synthesis. Graham *et al.* 2016 address the question "When do we need to accurately predict microbial community structure to accurately predict function?" In this re-analysis of 82 existing datasets of bacterial community structure and a variety

of ecosystem processes (both C and N cycling) the authors show that microbial community metrics had low power to explain ecosystem process rates but they improved models based on environmental variables alone by on average 8%, which while significant is admittedly not stellar.

In particular, they found that models based on all predictor sets (environmental variables only, microbial parameters only, or environmental + microbial parameters) had very low power to explain denitrification rates but that community diversity metrics added more explanatory power for denitrification rates than for any other process (which partly justifies our approach). The aim of our study was to achieve a better understanding of the relationships between the structure, abundance, and activity of denitrifiers over a range of dairy-pasture soils. As justification of this aim we suggest that this 'may enhance our ability to promote complete denitrification in order to reduce $N_2O$ emissions from pastoral agriculture'.

Given the results of Graham *et al.* 2016 we concede that this now seems overly optimistic and we have revised the introduction to reflect this, however, we point out that the former study did not directly analyse $N_2O:N_2$ ratios during denitrification. We have also made a large number of revisions to refocus the manuscript on our central question which is 'if the size and activity of bacterial denitrifying communities can be predicted on the basis of soil physicochemical characteristics'. We feel that this is clearly a separate question than that addressed by Graham *et al*. but one that could shed additional light on the environmental contexts wherein microbial community structure and diversity can inform ecosystem function.

Many new molecular methods have been developed over the last decennia, opening possibilities to study the microbial life in soils. The impression is that the availability of a method designed this study. Results and conclusions are vague.

Author's Response: This is unfortunate and points clearly to the need for a thorough revision of our manuscript in order to better frame its goal. In response to this comment we have completely revised the results and discussion.

**Scientific quality: Are the scientific approach and applied methods valid? Are the results discussed in an appropriate and balanced way (consideration of related work, including appropriate references)?**

The authors are familiar with molecular and microbial genetic and process studies, which were applied here. However one can ask what can the denitrifier community structure tell on the $N_2O$ emission size?

Author's Response: Here we present qPCR data for the number of gene copies for the functional genes *nir*S, *nir*K and *nos*Z, as well as for the ratio of *nos*: *nir*. The ratio of these genes has been interpreted previously as an index of the potential for complete denitrification (Phillipot *et al.* 2011, Braker *et al.* 2012, Jones *et al.* 2014). Generally, it is expected that soils with high *nos*: *nir* ratios are more likely to emit proportionally smaller N as $N_2O$. We have now clarified this in the methods section.

A DEA assay gives a hint in combination with *nos*Z genes. But contrasting results were found, where soil of group had low DEA and low nosZ (Fig 3), so what to expect? And soil group 2 high in nosZ where DEA was the highest, does that hint low $N_2O$ in spite of high process rate?

Author's Response: Group 2 soils (based on soil physicochemical characteristics) varied widely with regard to both denitrification enzyme activity (DEA) and the number of *nos*Z gene copies (Fig 1 and Fig 3) but within a soil these parameters largely agreed. This agreement drove the significant positive correlation among DEA and *nos*Z copy numbers which we report in supplementary table S5. Both high DEA and higher *nos*Z gene copy might indicate low $N_2O$ despite high denitrification rate under most favourable condition in these soils. The revised discussion is substantially clearer on this point.

It is not possible to guess that $N_2O$ may be emitted from a soil. This is not discussed in the paper. However $N_2O$ emission size was not the main aim of the study, but the study was motivated by it. The motivation of the study is vague (see above), and the objective told in the abstract 'to gain insight to relationships between structure and activity'.

Author's Response: As above, we have rewritten the introduction section to deemphasise a direct link between denitrifier community size/structure and $N_2O$ emissions from soils.

What was the insight gained? Ten soils were compared, but one soil (n=1?) is treated as a group of soils (group 2), however many samples at one site. This could be questioned?

Author's Response: The soils grouped into 3 distinct clusters based on their physicochemical characteristics. This is a result, not an aspect of our sampling design. We then ask whether microbial community diversity, structure and size varied according to these same major gradients in

physicochemical characteristics. We find that they do not, but rather responded primarily to soil water content and Olsen P. This is much more clearly communicated in the revised manuscript.

References to papers describing methods are not appropriate, since the methods are not found there.

Author's Response: Thank you, we have replaced the erroneous reference with the correct one.

5   The Discussion section resembles a Result section however there are references after each paragraph.

Author's Response: As above, we have thoroughly revised the results and discussion sections.

**Presentation quality: Are the scientific results and conclusions presented in a clear, concise, and well-structured way (number and quality of figures/tables, appropriate use of English language)?**

The authors could have better worked the text through. Sometimes the text is difficult to follow. The

10  overall structure is OK, however the content of the discussion could couple more to other work.

Author's Response: Thank you, we have thoroughly revised the results and discussion sections.

**Specific comments**

P2 L34 This hypothesis is not very visible through the paper. Management practices altering environment conditions at the different soils could not be found.

15  Author's Response: Given the centrality of soil water content in driving bacterial denitrifier community metrics in our study we have modified the discussion section to include a more thorough discussion of the ways in which pasture management can influence soil water content.

P3 L6 'Population therefore' something lacking, difficult to read.

Author's Response: Revised.

20  L17-20 This section describing soil sampling is messy, difficult to read, some things are lacking like only one soil depth here but two depths later on.

Author's Response: Additional information has been included to clarify the soil sampling.

L23 Standard protocols refers to Morales et al. (2015), but I could not find these methods referred to in this reference. L28 Refers to Morales also for DEA, not in that paper. I have to say I have not checked all

25  references given in the manuscript.

Author's Response: As above, we have replaced this erroneous reference with the correct one.

P4 L25 Why was the 10 soils investigated described so sparsely?

Author's Response: Detailed description of the 10 soils investigated has been provided in the supplementary section.

P5 L32 Two soils (n=2) compose one group. Enough?

P6 L2 More so for group 2 consisting only one soil.

Author's Response: As above, the soils grouped into 3 distinct clusters based on their physicochemical characteristics. This is a result, not an aspect of our sampling design.

P7 L12 two orders of magnitude? Only one as I can see.

Author's Response: Thank you, this was a typo that has been corrected.

Many vague and not very clear statements and conclusions, based on one or two soils follows.

Author's Response: We have thoroughly revised the manuscript to avoid any vague or unclear statement.

**Anonymous Referee #2**

 1)**

**Scientific significance: Does the manuscript represent a substantial contribution to scientific progress within the scope of Biogeosciences (substantial new concepts, ideas, methods, or data)?**

The manuscript is aiming at unravelling the relationships between denitrifier community structure and environmental parameters in pasture soils. It is well within the focus of the journal. The methods used are solid but not cutting edge and suited to answer some of the questions. However, the experimental design is not perfect for the big aim of understanding the connections between nitrous oxide emissions, denitrifier community structure composition and soil type and land management.

Author's Response: As in our responses to R1 above, we concede that our aim of understanding the link between the structure, abundance, and activity of denitrifiers based on soil physicochemical characteristics may not directly 'enhance our ability to promote complete denitrification in order to reduce $N_2O$ emissions from pastoral agriculture' and we have now revised the introduction to reflect this.

**Scientific quality: Are the scientific approach and applied methods valid? Are the results discussed in an appropriate and balanced way (consideration of related work, including appropriate references)?**

In principal I think the study has great potential but in present form suffers a little from too many variables between the different soils and not enough samples/replicates of similar soils to resolve their influences.

Author's Response: We present n=6 for all soil physicochemical datasets and n=3 for molecular microbial datasets. However molecular work was based on 6 separate DNA extractions followed by pooling 2 extractions/PCR amplification in attempt to better represent potential spatial variability among replicates.

I further have a slight problem with the determination of copy numbers for functional genes and using these numbers as 'abundances' of the organisms. The denitrifiers could be the same percentage of the total population in all soils and it would make sense to at least also determine the copy numbers of the bacterial 16S rRNA gene with a general primer set. Then there are still issues with gene copy number per genome, functional gene/16S rRNA gene ratio in a genome and such left, which would be harder to account for.

Author's Response: Yes, agreed. This problem is inherent in many qPCR studies of functional genes. We have revised the methods and results sections to reflect this limitation of our approach. In particular, we have moved figure 4a to the supplementary data so that our results and discussion focus on the *nos*: *nir* ratio only. Because these genes do not always (but can) co-occur within an organism their ratio may better reflect cell numbers of complete: incomplete denitrifiers. Of course, this assumes similar PCR bias among the different primer sets, but that assumption applies equally to amplification of a "housekeeper gene" like 16S rRNA or *rpo*B.

From an organismic point of view it has to be considered that the nirS/K and nosZ genes are not distributed completely independent. They are linked in organisms that can perform the full denitrification pathway. Therefore it is quite surprising that the NMS analysis of nosZ (Fig. 3) doesn't show any clustering while nirS/K did. Would it be possible to identify the T-RFs of nirS/K that have similar distribution patterns over the samples than those from nosZ? That way only subsets of T-RFs could be analyzed in order to determine how the soil parameters influence their presence/abundance.

Author's Response: This is an interesting suggestion. It would certainly shed light on the how complete denitrifiers respond to varied soil conditions. However, this is really a separate question from the one we pose here because complete denitrifiers are typically only a small subset (~ 0.5%; Deslippe *et al*. 2014) of all denitrifiers in New Zealand pasture soils. Should we follow this suggestion, we would miss incomplete denitrifiers, which are equally likely to be affected by the soil physicochemical characteristics we study here, and they are especially of concern for GHG emissions.

The discussion is a bit lackluster and is missing a part in which the results are discussed in the frame of the bigger question, nitrous oxide emissions. Especially as the results of the study seem to suggest that all the soil parameters collected do not explain the distribution and abundance of the nosZ gene over the different soils. How does this fit with the question? I would have expected a more thorough discussion of this, also the potential pitfalls of the methods used that could have influenced this result (primer bias, etc.).

Author's Response: Yes agreed. We have thoroughly revised the discussion section and we now more fully address reasons that the distribution and abundance of *nos*Z genes respond primarily to SWC and Olsen P in our study.

**Presentation quality: Are the scientific results and conclusions presented in a clear, concise, and well-structured way (number and quality of figures/tables, appropriate use of English language)?**

The quality of the presentation is lacking a little with sentences that sometimes need re-reading before they make sense. Minor grammar mistakes here and there can be found too as well as layout issues.

Author's Response: We have given the manuscript a general overhaul and respond to specific issues in detail below.

The figures are not always as informative as they could be.

Figure 1 doesn't resolve the differences between the sites closely located next to each other well. It gives a general impression where the sites are located but why not move it to SOM and then add three zoomed in insert maps that resolve the three local areas where the samples were taken better?

Author's Response: Thanks we have revised this figure and moved it to supplementary section (Fig S1).

Figure 2 is really busy, especially with the legend for each dot. As the color code already defines which sampling site they are from, why not just put the numbers for the replicates on? And I don't think it adds anything to know which exact replicates are closer together as it is not mentioned elsewhere in the manuscript. So it might be an idea to leave the annotations in the figure off altogether and just rely on the color code explained in the legend. Further, the circles defining the clusters should not cross the borders of the ordination.

Author's Response: Done. Good suggestion, thanks. We have made the changes to this figure as suggested (Fig.1).

Figure 3 is again pretty busy and would need some cleaning up. It would also make sense to stick to the same symbols/colors as in Fig. 1. Fig. 3 is pretty meaningless as the majority of samples can't be resolved in the presented ordination. Here the question is if an outlier analysis could be used to remove the data points at the edges of the ordination. If not, I would suggest to at least show an ordination with only the data points that cluster tightly together in the SOM to resolve potential trends in this subset of samples that is not affected by the 'outliers'.

Author's Response: In this version of the manuscript we have recreated figure 3. It now retains the same symbol colours as in figure 2, but has different symbol shapes to communicate the soil groups (based on the PCA result). We disagree that *nir*S ordination is meaningless because it illustrates that *nir*K community structure responded to the same physicochemical characteristics (SWC and Olsen P) as *nir*S communities did, which is a major point of the manuscript. However we acknowledge that the importance of this result was not sufficiently described in the previous results section nor was it adequately discussed. Consequently, in this version of the manuscript we have corrected those issues as well. While we disagree that outlier analysis is appropriate in this case (removal of HR and PL soils constitutes a 20% data reduction), as requested we have, added an ordination of the *nir*K data without PL and HR soils to the supplementary materials, which shows that Olsen P and soil water variables remain the primary driver of *nir*K community structure, even for this reduced dataset. Likewise, we have added this information to the results and discussion.

The data presented in table 2 would also make a nice figure, maybe even in combination with Fig. 4.

Author's Response: Agreed. Since the patterns of significance were similar for gene richness, evenness and diversity we chose, (for the sake of simplicity) to make a figure (fig. 2) illustrating only gene richness by soil group. We have moved table 2 to the supplementary section.

**Specific comments**

Multiple pages: gene names are normally all italicized, also e.g. the 'K' from 'nirK'

Author's Response: Thanks, we have now thoroughly checked the manuscript for italicized gene name.

p 3, l 16: Sampling was conducted between August and December. Where there any kind of controls to test for seasonality effects?

Author's Response: Our aim was to sample from the range of soil conditions that occur on NZ pasture farms. It was therefore important to sample in both wet and dry seasons. However it was not our intention to characterise the amplitude of seasonal variation within any given soil, and so we did not design controls that would allow us to assess seasonal variation. However, to ensure that our sampling spanned the range of soil moistures that are typical for pasture soils in NZ we sampled the soils that were expected to be wettest (OH and TeK) in winter and the soils that were expected to be driest in summer (PS, LM MF) the other soils were sampled in between these. We have clarified this in the methods section.

p 3, l 18: Were the 25 soil cores per replicate homogenized and mixed during the process of sieving?

Author's Response: Yes. Thanks for pointing out that this was unclear. This information has been added to the methods section.

p 3, l 18: Were all samples besides the ones for molecular data stored at 4 ∘C? If some of the analyses were done 6 months later I would be worried about changes in the soils as microbial activity will continue, although much slower.

Author's Response: The soils used in this study were collected over a nearly 6-month period. After each soil was sampled it was immediately sieved and pH, nitrate ($NO_3^-$) & ammonium ($NH_4^+$) –N (mineral-N), total nitrogen (TN), total carbon (TC), Olsen phosphorus (P), microbial biomass carbon (MBC), soluble C, and denitrification enzyme activity (DEA) were measured within 1-2 weeks. Measurements of DEA and MBC were prioritized so that they typically occurred within the first few days after sieving. However, after the first two sets of soils were sampled, a technical problem with our analytical set-up caused delay in measurements of MBC for nearly 3 months. Given that it was not possible to go back to farms and resample all soils (as their physicochemical properties were likely to have changed in this time, we remeasured MBC on the initially sampled and stored soils after 3, 4 and 7 months, we determined that no significant changes in MBC occurred between the time period of 3 and 7 months. We therefore report MBC data for all soils that were stored between 4 and 6 months. We understand that this issue can be confusing to readers so we have clarified this in the methods section, as simply as possible.

p 6, l 8 ff/table 2/figure 3: The number of T-RFs used for the NMS analysis seems to be quite low and in the case of nirK also pretty different between the samples. This could result in problems with the

ordination that is hard to evaluate. It would be nice to report stress values and also provide the data matrices used for the NMS analysis in the SOM so the reader can evaluate them.

Author's Response: Thank you for this useful comment. Total T-RF richness was $nir$S=52, $nir$K=53, $nos$Z=47, which is quite typical for T-RFLP studies of functional genes. However, you are correct that the minimum and maximum number of T-RF varied among samples, which could possibly have contributed to instability in the NMDS ordinations we present. Final stress for the three ordinations in fig 3 were as follows: $nir$S=12.5, $nir$K=5.5, $nos$Z=9.4. So this was clearly not a major problem in our datasets. Nevertheless this is a good point and we have added this information to the discussion and SOM sections. We would also point out that the new T-RF richness figure (and specifically the size of the error bars on the histograms), which we have produced in response to your earlier comment, will also help our readers to evaluate variability in gene richness among samples in our study.

p 8, l 14: Wouldn't it have been possible to avoid uneven grazing and excretal deposition by fencing off an area a couple of weeks prior to sampling? Or at least try to avoid these spots by a careful screening of the area to find representative spots?

Author's Response: All of the pastures sampled in this study were fenced from livestock and none had been grazed within 8 weeks of sampling. Thank you for pointing out this omission; this has now been added to the methods section. As explained in the methods section we also avoided any (old) dung patches when sampling, as bovine gut bacteria could have contaminated the soil sample if we had pushed the soil corer through a dung pile, and so we did not do this.

p 9, l 10 ff: I am not sure why the authors are so surprised by this. The sampling procedure (25 cores combined) should diminish the signals from different microniches and create an integrated signal.

Author's Response: True, but as we say we would then expect $nir$S and $nir$K to be negatively correlated overall. No significant negative correlation between $nir$S and $nir$K suggests independent environmental or stochastic controls on the size of these populations. This section has been expanded in the revised discussion.

p 10, l 21: 'saturated': I assume with water?

Author's Response: Yes, clarified.

p 10, l 24 ff: If the adsorption of copper is the reason that there is less nitrous oxide reduction, then why are there active nirKs, which also have copper as co-factor? There must be another explanation for this observation or could a reduction in the copy numbers of nirK be observed in these soils as well?

Author's Response: Yes, thanks for pointing out that our argument was confusing. We have revised the conclusions to make the point clearer. We did not intend to suggest that adsorption of copper is the reason that there is less nitrous oxide reduction in allophanic soils, but rather less nitrite reduction. We agree that because allophanic soils adsorb copper, they are likely to select against $nir$K denitrifiers. We expected this to reduce the overall number of genes encoding nitrite reductase in group 1 soils, but we didn't observe this (Fig 4). We have revised the discussion section of the manuscript to include this point. The point of interest in the conclusion section is that, the $nos$:$nir$ gene ratio data we show agrees with previous work by our group showing that allophanic soils emit greater $N_2O$: $(N_2O + N_2O)$ relative to other soil types.

**Anonymous Referee #3**

 **Summary:**

They sampled soils from 10 different geographical locations in New Zealand. They did an ordination of soil characteristics and found that the 10 sample locations could be grouped into 3 groups based on soil characteristics. These groupings were used in the further analysis of T-RFLP, qPCR and DEA data.

**General comments:**

The study attempts to find how various pasture management (soil water, carbon and fertility) will affect the denitrifier community, which increase our knowledge on denitrification in different soil types, and maybe improve our ability to promote complete denitrification and avoid N2O emission. This is a relevant question within the scope of BG. They find that fertile soil with high microbial biomass promote complete denitrification, whereas allophanic saturated soil is a source of N2O production.

I found it hard to get a good overview of the results and discussion, maybe because of poor flow and clarity in writing. I agree with RC1 that the discussion resembles a result section. In general every section sums up observations and have some explanation with a reference. I don't think it reaches a high enough

level of discussion. I'm also not confident that the data is strong enough to answer the question sufficiently. qPCR on RNA would be more reliable. To my knowledge the nir genes are very ubiquitous and not necessarily expressed.

Author's Response: We agree with R3's assessment that these doubts stem from poor flow of the manuscript and a lack of clarity in writing. These comments align with those of the other reviewers and made it clear to us that the manuscript required a major overhaul. To this end we have revised all parts of our manuscript as detailed elsewhere. Now that this is done we feel that our aim of achieving a better understanding of how soil physicochemical characteristics' affect the size, structure and activity of bacterial denitrifying communities is clear, and we think that R3 would agree that qPCR of RNA would not be an appropriate tool with which to address it.

Both title and abstract are descriptive and clear, reflecting the study well.

**Specific comments:**

The whole introduction argumentation for this study (P2, L11 – P3, L2) makes a good background, but somehow it's a bit vague. The idea of the study is very good and this framework can make it more visual with clearer and stronger formulations.

Author's Response: To this end we have added to the introduction one sentence, immediately after the statement of aim: "In particular, we asked if the size and activity of bacterial denitrifiers could be predicted on the basis of soil physicochemical characteristics."

P3, L22-23 I would mention which physicochemical characteristics were used in this study here, otherwise you only see it when reading the statistical analysis.

Author's Response: Added, thank you.

Regarding methods for physicochemical characteristics, DEA and qPCR, they refer to Morales et al. (2015). This seems to be another study of the very same soil sampling, and this manuscript is reusing data from Morales et al. (2015), right? It should appear more clearly that this study is an extension of Morales et al. (2015) with reuse of data. It would also seem natural to refer more to the earlier study since it's the same topic. There should be references to this in the introduction and/or discussion, not only for methods description.

Author's Response: Yes that's right, some of the physicochemical and molecular data presented here also appears in Morales *et al*. (2015), although the data analysis and objective of both the studies is entirely different. We revised the methods to more clearly convey that point, we also now refer to the Morales *et al*. (2015) in introduction of our paper.

5   P10, L25-29 Suddenly in the end of the conclusion this new stuff about allophanic soils comes up, this should have been included earlier on. The conclusion should instead round and wrap up. New stuff should not be introduced like this.

Author's Response: Yes, we agree and include the point about $N_2O$ emissions from allophanic soils in the discussion too.

10  **Technical corrections:**

Inconsistent use of water content terms and abbreviations: "Moisture"/"soil water"/"soil water content"/"SWC" and also "% SWC at field capacity"/"% FC SWC"/"high moisture at FC". Also "Field fresh" (P3, L20) and "field-moist" (P3, L22). This was all quite confusing to me.

Author's Response: Thank you, we have revised all parts of the manuscript with an eye for consistency.

15  Figure 2 have too many abbreviations in caption, the figure itself should be more descriptive.

Author's Response: This same comment was made by R2 and so we have changed Fig 2 accordingly.

In caption for Figure 4, SEM should first be defined and then used. Not the other way around.

Author's Response: Agreed, done that.

P1, L3 There should not be a dot in the end of the manuscript title. This also occurs in the titles in the

20  references.

Author's Response: We have removed dot from the end of the manuscript title and also from the titles in the references.

P2, L34 With enhanced structure, do you then mean diversity?

Author's Response: This comment has been rephrased for clarity.

25  P3, L19 "2 depths" not "2 depth". I can't find which depths you chose (mm/cm?), should be stated in the methods.

Author's Response: Yes thanks, we have fixed this and also added the unit of measurement.

P4, L7-8 "2.5 ul of 10xPCR buffer (1 mM MgCl2), 0.5 mM MgCl2". Final concentrations in reaction mix should be stated, this looks weird to me.

Author's Response: Okay, we have rewritten as final molarity.

P4, L24 I would specify that the qPCR was performed on DNA

Author's Response: The title of the section "Quantitative polymerase chain reaction (qPCR) of total bacterial and denitrifier genes" makes this point clear.

P5, L19 Isn't the right abbreviation NMDS? Not NMS

Author's Response: Both abbreviations are in common use, with variation stemming from the term used by the particular stats package. PCOrd software refers to NMS ordinations (McCune and Grace, 2002), thus our use of that abbreviation here.

[revised manuscript text omitted]
 moisture and that soil physical properties or and that pasture management practices that alter increase soil water, would carbon and fertility will enhance the size and , structure and activity of the denitrifiers community. Knowledge about denitrifier activity under contrasting conditions may enhance our ability to promote complete denitrification in order to reduce $N_2O$ emissions from pastoral agriculture.

**2 Materials and Methods**

**2.1 Sites and soils**

Our aim was to sample soils that would encompass the range of physiocochmical conditions that predominate on New Zealand dairy farms. We therefore targeted The soils were selected on the basis of their geographical location (North or South Island of New Zealand), and variation in mineralogy (allophanic or non-allophanic soils). As soil moisture is a key factor affecting the structure and activity of soil denitrifier communities (Liu et al., 2012; Mergel et al., 2001) it was also important to sample in

both wet and dry seasons. We therefore sampled soils over a 6-month period from winter to summer. Soil textures varied from a stony silt loam to a fine sandy loam, and the sites ranged from poorly drained to well drained (Table 1). We sampled soils expected to have the greatest soil water contents in winter and those we expect to be driest in summer, with other soils sampled in between these times (see supplementary table 1X for soil sampling dates). Our objective in this study was to understand the influence of variability in soil properties on denitrification activity and denitrifier population therefore, pasture soils with varying physical and chemical characteristics were We collected soils from 10 different New Zealand dairy farms (Fig. 1). All the sites New Zealand commercial dairy farms (Fig. S1). All sites were fenced from livestock and none had been grazed within 8 weeks of sampling. All sites the from which the soil samples were collected are commercially managed grazed dairy pastures were dominated by perennial ryegrass (*Lolium perenne)* and white clover (*Trifolium repens*). The soils were selected on the basis of their geographical location (North or South Island of New Zealand), and variation in mineralogy (allophanic or non-allophanic soils). Soil textures varied from a stony silt loam to a fine sandy loam, and the sites ranged from poorly drained to well drained (Table 1). Fertilization regimes varied among the farms and consisted of applications of 150–200 kg N ha$^{-1}$ annually. Detailed descriptions of the individual fertiliser applications at the 10 farms are described in the supplementary information.

*Insert Fig. 1*

Insert Table 1

**2.2. Sampling and analysis of soil properties**

At each For the collection of soil samples, on each farm, we randomly selected six random locationblocks of 100 m$^2$ area were identifiedfor the collection of soil samples. At randomly selected locations within each block, tSoil samples were collected between August and December 2010 once from each site. Twenty–five soil cores (25 mm diameter × 100 mm long) were collected obtained from the 0–100 mm depth using a steel corer. The 25 cores from each block were pooled to form a single composited sample per block (*n* = 6 composited soil samples per farm). The 25 cores from each location were pooled, but the 6 replicates from each farm All swere stored separately (*n* = 6). Soil samples were collected between August and December 2010 once from each site. Soil cores samples were taken to the laboratory,

individually  homogenised, sieved to 2 mm and stored at 4ºC in plastic bags (10 sites × 6 replicates = 60 samples). ~~All of the pastures sampled in this study were fenced from livestock and none had been grazed within 8 weeks of sampling. Twenty-five soil cores (25 mm diameter × 100 mm long) were collected from the 0-100 mm depth using a steel corer from six random locations of 100 m² area on each farm (once) between August and December 2010 (10 sites × 6 replicates = 60 samples). The 25 cores from each location were pooled but the 6 replicates from each farm were stored separately (n = 6). Field-fresh soil cores were taken to the laboratory, sieved to 2 mm, and stored at 4ºC in plastic bags.The~~ We measured pH, nitrate ($NO_3^-$) & ammonium ($NH_4^+$) –N (mineral-N), total nitrogen (TN), total carbon (TC), Olsen phosphorus (P), and soluble C, on the field-moist sieved soils using standard protocols (for details see Jha 2016). Soils were analysed for these parameters within 2 weeks of sampling.

DEA was determined using  the acetylene inhibition method described in Luo et al. (1999), with the exception that we added chloramphenicol to inhibit the *de novo* synthesis of enzymes. Thus the values we report represent only the existing enzyme activity in soils. DEA was assessed for all soil samples within 2 days of collection. DEA incubation conditions and the  gas sampling methods are  described elsewhere (Morales et al., 2015). We intended to measure microbial biomass carbon (MBC) within 48 hours of soil sampling but a technical problem with our  set-up delayed measurements of MBC for nearly 3 months. To standardize this effect across soil samples we monitored changes in the size of the MBC pool in two soils over 7 months. We found that no significant changes in MBC occurred between 3 and 7 months for soils stored at 4ºC (see supplementary table 1). We therefore report MBC data for all soils that were stored at 4ºC between 4 and 6 months.

**2.3 DNA extraction from soils**

Within six months of soil sample collection, soil samples were thawed on ice and a  0.25 g aliquot was obtained. DNA was extracted from these soil samples using the MoBio PowerSoil™ DNA Isolation Kit (MoBio, Solana Beach, CA, USA) following the manufacturer's instructions. The yield and quality of DNA extracts were verified as described in Deslippe et al. (2014).

5  DNA was stored at –20°C until analysed.

**2.4 Terminal restriction fragment length polymorphism (T–RFLP) of denitrifier genes**

Terminal restriction fragment length polymorphism (T–RFLP) was performed to analyse the community structure and diversity of  *nir* and *nos* genes in soil samples. T-RFLP for *nir*S and *nos*Z

10  genes was conducted as described in Deslippe et al. (2014) except that PCR  for *nir* genes occurred in a total volume of 25 µl reaction mixture, which

 contained 2.5 µl of 10×PCR buffer (1 mM MgCl$_2$), 0.5 mM MgCl$_2$, 0.2 mM each deoxynucleotide triphosphate (dNTP), 1.25 U of *Taq* polymerase (Fisher *Taq*, Thermofisher Scientific® Inc.), 0.8 mg/ml Bovine Serum

15  Albumin (BSA), 1.0 µM of each primer, and 10 ng DNA template per reaction. The PCR amplification consisted of an initial denaturation of the DNA template at 94°C for 30 s, followed by 35 cycles of 20 s at 94°C, 20 s at 56°C, and 20 s at 68°C. The reaction was completed by 10 min at 68°C.

For T-RFLP of the *nir*K gene we used the primers Copper 583F, 909R (Dandie et al., 2011). The amplifications of *nir*K and *nos*Z genes were achieved under slightly different condition than the *nir*S gene

20  according to the specifications of the reagents used for PCR. The PCR amplification was performed in a total volume of 25-µl reaction mixture containing 10 µl of 2 × NEB Taq master mixes (New England Biolabs® Inc.), 0.4 µM of each primer, and 10 ng DNA template per reaction. PCR consisted of an initial denaturation of the DNA template at 94°C for 2 min, followed by 35 cycles of 30 s at 94°C, 1 min at 56°C, and 1 min at 72°C. The reaction was considered complete after 10 min at 72°C.

25  The T-RFLP profiles generated for the soil samples were analysed using Peak Scanner® v1 software (Life Technologies) and as described in Deslippe et al. (2014). The total number of terminal restriction fragments (T-RFs) per electropherogramme was taken to indicate genotype richness per sample. We then

calculated the gene Shannon's diversity index and Pielou's evenness index (Magurran, 1988) per sample and used 1-way analysis of variance (ANOVA) to determine if soils belonging to the three physicochemical groups differed with respect to gene richness, evenness and diversity.

**2.5 Quantitative polymerase chain reaction (qPCR) of total bacterial and denitrifier genes**

5   Quantification of bacterial *nir*S, *nir*K, and *nos*Z genes was accomplished using qPCR, following the methodology of Deslippe et al. (2014), . Amplification efficiencies of qPCR reactions for samples were within the  range of values (E = 90–110%) published previously (McPherson and Moller, 2006) . The reactions were linear over 7 orders of magnitude and sensitive down to $10^2$ copies. The ratio of abundances of

10  denitrifier genes in environmental samples has been interpreted previously as an index of the potential for complete denitrification (Philippot et al., 2009). Here, we calculated the nosZ: (*nir*S+*nir*K),  of soil samples. We expected that soils with low nosZ: (*nir*S+*nir*K),  are more likely to emit a greater $N_2O:(N_2O + N_2)$.

15  **2.6 Statistical Analysis**

The normality and homoscedascity of allsoil physicochemical,  gaseous emissions, and biological datasets were examined using Anderson-Darling (Stephens, 1986) and Levene's tests, respectively in Minitab® 16 software (Minitab Inc.).

20  Box-Cox transformations (Box and Cox, 1964) were applied to data sets as required to conform to model.

25

normality of data were violated for some of the parameters, the data sets were transformed to normal using the Box-Cox transformations (Box and Cox, 1964).

The differences in the means of soil characteristics such as pH, nitrate ($NO_3^-$) & ammonium ($NH_4^+$) –N (mineral-N), total nitrogen (TN), total carbon (TC), Olsen phosphorus (P), microbial biomass carbon (MBC), soluble C, DEA, number of gene T–RFs and gene copy numbers were assessed using a one-way analysis of variance (ANOVA) test with soil type as a factor. Tukey's Studentized Range Test at $\alpha = 0.05$ significance level was used *post hoc* to reveal significant differences among means. The relationships among the soil chemical characteristics pH, nitrate ($NO_3^-$) & ammonium ($NH_4^+$) –N, TN, TC, Olsen P, MBC, DEA, number of denitrifier gene T–RFs, and gene copy numbers were determined using Pearson's correlation analysis.

In order to reduce the dimensionality of the many correlated soil physicochemical characteristics we performed principal components analysis (PCA). We included % soil water content (SWC), % SWC at field capacity (% FC SWC), pH, TN, TC, Soluble C, Olsen P, nitrate ($NO_3^-$) and ammonium ($NH_4^+$) –N as factors in the PCA. Soils grouped along the first and second ordination axes. We used multiple response permutation procedure (MRPP) to assess the statistical significance of these groupings. MRPP calculates the chance-corrected within-group agreement (A), a measure of within-group homogeneity compared with that expected by chance, where A = 1 corresponds to identical members within each given group (maximum effect of factor), and where A ≤ 0 corresponds to within-group heterogeneity equal to or larger than that expected by chance (no effect of factor; McCune and Medford 1999). We also calculated Pearson correlations among soil microbial characteristics and the ordination axes, and plotted those that were significantly correlated (tau>0.2) with axis 1 and 2 as vectors on the PCA. In order to assess how denitrification and soil microbial properties varied with the physicochemical characteristic of soils, we calculated Pearson and Kendall correlations among physicochemical characteristics and the ordination axes, and plotted those that were significantly correlated with axis 1 and 2 as vectors on the PCA. A nonparametric Kruskal Wallis test was performed to examine the differences in soil parameters among various group of soils obtained through PCA.

[revised manuscript text omitted]

Ordination of the soil samples in *nos*Z T-RF space revealed little clustering of soil samples by origin or group. Likewise, we detected no significant patterns of correlation among the first and second ordination axes and the soil physiochemical characteristics. However axis 1 of the NMS ordination, in 30.3% of the variation in *nos*Z community structure, that corresponded to the physicochemical characteristics of the soils (Fig. 3c).~~

*Insert Figure 3*

**3.2.3 Denitrifier gene abundance**

The  number of *nir*S and *nir*K gene copies varied  widely among the 10 soils; *nir*S gene copies ranged from $2.5 \times 10^7$ to $3.9 \times 10^8$ copies g$^{-1}$ soil, while *nir*K gene copies varied from $2.3 \times 10^8$ to $5.9 \times 10^8$ g$^{-1}$ soil . Overall soils, *nir* genes

5  were on average an order of magnitude more abundant than those encoding the final step of denitrification. overtwo orders of magnitude among the soils in three groups3~~a), with the effluent irrigated

10 (group 2) soil having intermediate values. Despite large variability, t the group 2 soil had  significantly more *nosZ* gene copies than the soils in group 1 and 3.  while the group 1 soil had significantly fewer than the other two groups (P<0.005, Fig. 4a).  Consequently the ratio of *nosZ*:*nir*S+*K* genes, which may indicate of the relative

15 abundance of complete denitrifiers, varied significantly among the three groups of soils (Fig. 4b), with the  effluent irrigated group 2 soil harbouring the highest, the allophanic soils of group 1 the lowest and the group 3 soils intermediate ratios of *nos:nir* genes.

20 ~~The abundance of *nir* and *nos* genes also varied with the major physicochemical characteristics of the soils. The allophanic soils of group 1 were similar in harbouring relatively large *nir*S+*K* communities but were among the smallest *nosZ* communities. Consequently these soils had similar and low *nos:nir* ratios, with complete denitrifiers comprising approximately 1% of all bacterial denitrifiers. The group 2 effluent irrigated soil had relatively large *nir* and *nos*~~

25

 *Insert Figure 4*

**3.3 Denitrification enzyme activity (DEA)**

DEA varied considerably among the pasture soils, but also among replicates  within a soil (Table S3). DEA

varied by a factor of five among the soil groups, with the group 2 soil ( effluent irrigated (MWEI) achieving significantly higher DEA than other groups, while the soils of  group 1 (HR & OH) had significantly lower DEA values than other groups (H = 12.09, $P$ = 0.02; Table S3). DEA varied considerably in soils belonging to group 3. Overall, DEA was mostly strongly positively correlated soil $NO_3^-$–N contents and was mostly strongly negatively correlated to soil $NH_4^+$–N contents, driving its significant correlation with axis 2 of our PCA (Fig 3). All significant physicochemical  correlates to DEA are given in Table S5.

was mostly strongly soil $NO_3^-$–N content mostly strongly negatively correlated to soil $NH_4^+$–N contents, driving its significant correlation with axis 2 of our PCA (Fig 2).

3.4 Relationships among denitrification and denitrifier community size and structure across a range of soil moistures

Given that the structure of *nir*S, *nir*K and *nos*Z communities varied primarily in response to soil water content and Olsen P (Fig 3), we wished to know if unique relationships between the richness and size of the denitrifier gene community and DEA exist at different SWCs.  To address this question, we categorized soils according to coarse-scale SWC (high, moderate and low) and examined Pearson's correlations among these variables within soil SWC categories.

determine the relationship between DEA, and denitrifier gene abundance and richness across a range of SWC. This revealedF that or in soils in the highest with high SWC category (MWEI, OH, and HR), we found that strong and significant positive correlations existed between denitrifier gene copy numbers and DEA was significantly correlated to their denitrifier gene abundance [*nos*Z (r = 0.643, $P$ = 0.049), *nir*K

5 (r = 0.821, $P$ = 0.007)], and *nir*S (r = −0.887, $P$ = 0.001)] and . Likewise, strong and significant positive correlations existed between DEA and thedenitrifier gene T-RF richness of denitrifier genes [*nos*Z (r = 0.801, $P$ = 0.010), *nir*K (r = 0.783, $P$ = 0.013), and *nir*S (r = 0.793, $P$ = 0.011)]. However, these patterns of correlation were not present in the soils categorized as moderate or low SWC.

In particular, the group 1 soil (MWEI), which had the highest Olsen P, microbial biomass, soil $NO_3^-$–N

10 content, *nos*Z gene copy abundance, and *nos*Z gene phylotypes richness, diversity, and evenness also had higher DEA of other two groups.

[revised manuscript text omitted]

**Figure 2: Ordinance of Soil Characteristics in 1st and 2nd PC axes (0–100 mm depth). MWEI = Manawatu fine sandy loam (effluent irrigated); MW = Manawatu fine sandy loam; TM = Tokomaru silt loam; TeK = Te Kowhai silt loam; OH = Otorohanga silt loam; HR = Horotiu silt loam; PS = Paparua silt loam (Springston); LM = Lismore stony silt loam; MF = Mayfield silt loam; PL = Paparua silt loam (Lincoln). NO₃⁻ = Nitrate-N content, NH₄⁺ = Ammonical –N content, MBC = Microbial biomass content, FC = Field capacity. Numbers adjacent to soil codes represent replicate number. Numbers in bold inside the ovals indicate group numbers.**

[Figure]

[Figure]

**Fig. 1** Ordination of Soil Characteristics

[Figure]

**Fig. 2** Gene T-RF richness in different soil groups. White, grey and black columns denote groups I, II and III soils. Mean values are reported ± 1 standard error of the mean. Columns with the same letters are not significantly different.

[Figure]

**Fig. 3** Non-metric multidimensional scaling (NMS) ordinations illustrating the Bray-Curtis dissimilarities of *nir*S, *nir*K and *nos*Z communities. Vectors represent those factors that were significantly correlated to the first and second ordination axes at tau = 0.2. Symbol colours are the same as in Fig. 2. Symbol shapes represent the three groups of soils as determined by PCA of their physiochemical characteristics (Fig 1).

[Figure]

**(a)**

**(b)**

Fig. 4 (a) Denitrifier gene (*nir*S, *nir*K, *nos*Z) copy numbers in different soil groups, error bars denote S.E.M. (b) Denitrifier gene abundance ratio (*nos*Z : *nir*S, *nir*K) in different soils groups. Mean values are reported ± 1standard error of the mean. Columns with the same letters are not significantly different. Letter values with same case or font denote one test (one test for each of the genes). Group 1 soils -OH and HR, group 2 soils -MWEI, group 3 soils – MW, TeK, TM, PL, PS, LM, MF. Abbreviations of soils are described in the Fig. 2 legend

**Table 1: Description of soils**

| Soil | Location of the dairy farm | Geographical Location | Soil Abbreviation | Soil Classification | Mineralogy Class |  |
|---|---|---|---|---|---|---|
| Te Kowhai Silt Loam | AgResearch Ruakura, Waikato | 37°44'57.55"S 175°10'27.06"E | TeK | Typic Orthic Gley | GlassyVolcanic , Kaolinitic |  |
| Otorohanga Silt Loam | Tokanui, Waikato | 38°11'19.70"S 175°12'35.67"E | OH | Typic Orthic | Allophanic |  |
| Horotiu Silt Loam | AgResearch Ruakura, Waikato | 37°46'30.80"S 175°18'23.27"E | HR | Typic Orthic Allophanic | Allophanic |  |
| Tokomaru Silt Loam | Massey University, Palmerston North | 40°22'58.50"S 175°36'31.01"E | TM | Argillic-fragic Perch-gley Pallic | Vermiculitic |  |
| Manawatu Fine Sandy Loam | Longburn, Palmerston North | 40°22'56.99"S 175°32'24.49"E | MW | Weathered fluvial recent | Illitic |  |
| Manawatu Fine Sandy Loam (Effluent irrigated) | Longburn, Palmerston North | 40°22'58.26"S 175°32'21.65"E | MWEI | Weathered fluvial recent | Illitic |  |
| Paparua Silt Loam (Springston) | Springston, Christchurch | 43°38'15.97"S 172°28'13.81"E | PS | Weathered Orthic recent | Illitic |  |
| Paparua Silt Loam (Lincoln) | Lincoln, Christchurch | 43°38'43.91"S 172°25'21.86"E | PL | Weathered Orthic recent | Illitic |  |
| Lismore Stony Silt Loam | Ashburton, Canterbury | 43°53'17.44"S 171°38'28.43"E | LM | Pallic Orthic Brown | Vermiculitic |  |
| Mayfield Deep Silt Loam | Methven, Canterbury | 43°38'30.12"S 171°43'47.28"E | MF | No data | No data |  |

Table 2: Richness, Pielou's evenness index, and Shannon's diversity index and of denitrifier gene terminal restriction fragments (T-RFs) in soils

| Group | Richness | | | Pielou's Evenness Index | | | Shannon's Diversity Index | | |
|---|---|---|---|---|---|---|---|---|---|
| | *nir*S | *nir*K | *nos*Z | *nir*S | *nir*K | *nos*Z | *nir*S | *nir*K | *nos*Z |
| 1 | $14.5 \pm 3.5^a$ | $3.5 \pm 0.9^c$ | $15.3 \pm 0.5^b$ | $0.6 \pm 0.1^a$ | $0.3 \pm 0.1^c$ | $0.7 \pm 0.0^b$ | $2.5 \pm 0.3^a$ | $1.1 \pm 0.3^c$ | $2.7 \pm 0.0^b$ |
| 2 | $15.0 \pm 0.6^a$ | $31.0 \pm 4.0^a$ | $26.7 \pm 1.2^a$ | $0.7 \pm 0.0^a$ | $0.9 \pm 0.0^a$ | $0.8 \pm 0.0^a$ | $2.7 \pm 0.0^a$ | $3.4 \pm 0.1^a$ | $3.3 \pm 0.1^a$ |
| 3 | $15.2 \pm 1.6^a$ | $11.8 \pm 1.8^b$ | $17.8 \pm 1.0^b$ | $0.7 \pm 0.0^a$ | $0.6 \pm 0.0^b$ | $0.7 \pm 0.0^b$ | $2.6 \pm 0.1^a$ | $2.2 \pm 0.1^b$ | $2.8 \pm 0.1^b$ |

Letters denote one way ANOVA test. Values sharing same letter are not significantly different in the column they are present in.
Where MWEI = Manawatu fine sandy loam (Effluent irrigated), HR = Horotiu silt loam, OH = Otorohanga silt loam, MF = Mayfield silt loam, MW = Manawatu fine sandy loam, TM = Tokomaru silt loam, TeK = Tekowhai silt loam, PS = Paparua silt loam (Springston), LM = Lismore stony silt loam, PL = Paparua silt loam (Lincoln); *nir*S and *nir*K = nitrite reductase gene, *nos*Z = nitrous oxide reductase gene.

---

## Author Response (AR2)

**Response to reviewers' comments:**

**Soil properties impacting denitrifier community size, structure and activity in New Zealand dairy-grazed pasture.**

Neha Jha, Surinder Saggar, Donna Giltrap, Russ Tillman, and Julie Deslippe

In this document we provide a comprehensive description of how we have responded to all the changes suggested by the associate editor.

**Anonymous Referee #1**

Comments

**Suggestions for revision or reasons for rejection (will be published if the paper is accepted for final publication)**

The authors have nicely improved their manuscript and make a much more compelling case. I have only two minor things:

Figure 3:
I agree with the authors that the final stress values for the NMDS ordination look normal. However, I still suggest reporting them in the figure legend to show the reader that this is not a problem as many will only read the final version of the manuscript and not go through the whole discussion.

Author's Response: Thank you for drawing our attention to this important information, we have reported the final stress values of all the three genes in the revised version of the manuscript

p. 10, l. 19: shouldn't this be anaerobic microsites?

Author's Response: Thank you we have corrected the typo. Changed aerobic to anaerobic microsites.